

# 16S rRNA gene metabarcoding reveals a potential metabolic role for intracellular bacteria in a major marine planktonic calcifier (Foraminifera).

Clare Bird*[1], Kate F. Darling[1,2], Ann D. Russell[3], Catherine V. Davis[3], Jennifer Fehrenbacher[3,4], Andrew Free[5], Michael Wyman[6], and Bryne T. Ngwenya[1]

[1]School of Geosciences, University of Edinburgh, Grant Institute, The King's Buildings, James Hutton Road, Edinburgh, EH9 3FE, UK
[2]School of Geography and Geosciences, University of St Andrews, North Street, St Andrews KY16 9AL, UK
[3]Earth and Planetary Sciences, University of California Davis, 2119 Earth and Physical Sciences, One Shields Avenue, Davis, CA 95616
[4]College of Earth, Ocean, and Atmospheric Sciences, Oregon State University, Corvallis, Oregon 97331
[5]School of Biological Sciences, University of Edinburgh, Roger Land Building, The King's Buildings, Alexander Crum
Brown Road, Edinburgh, EH9 3FF, UK
[6]Biological and Environmental Sciences, Faculty of Natural Sciences, Cottrell Building, University of Stirling, Stirling, FK9 4LA

*correspondence to: Clare Bird (clare.bird@ed.ac.uk)



**Abstract.** We investigated the possibility of bacterial symbiosis in *Globigerina bulloides*, a palaeoceanographically important, planktonic foraminifer. This marine protist is commonly used in investigations of climatically sensitive subpolar and temperate water masses and wind driven upwelling regions of the world's oceans. *G. bulloides* is unusual because it lacks the protist algal symbionts that are often found in other spinose species and has an atypical geochemical shell

signature. This is suggestive of a divergent ecology, making it a good candidate for investigating the potential for bacterial symbiosis as a contributory factor in shell calcification. Such ecological information is essential to evaluate fully the potential response of *G. bulloides* to ocean acidification and climate change. To investigate the ecological interactions between *G. bulloides* and bacterial populations in the water column, 18S rRNA gene sequencing, fluorescence microscopy, 16S rRNA gene metabarcoding and transmission electron microscopy (TEM) were performed on individual specimens of *G.*

*bulloides* (Type IId) collected from two locations in the California Current. Intracellular DNA extracted from five *G. bulloides* specimens was subjected to 16S rRNA gene metabarcoding and, remarkably, 37–87 % of all 16S rRNA gene sequences recovered were assigned to operational taxonomic units (OTUs) of the picocyanobacterium, *Synechococcus*. This finding was supported by TEM observations of intact *Synechococcus* cells in both the cytoplasm and vacuoles of *G. bulloides*. Concentrations were up to four orders of magnitude greater inside the foraminifera than those reported for the

Californian Current water column and approximately 5 % of the intracellular *Synechococcus* cells observed were dividing. This suggests that *Synechococcus* is an endobiont of *G. bulloides* Type IId, which is the first report of a bacterial endobiont in the planktonic foraminifera. We consider the potential roles of *Synechococcus* and *G. bulloides* within the relationship and the need to determine how widespread the possible symbiotic association is within the widely distributed *G. bulloides* morphospecies. The possible influence of *Synechococcus* respiration on *G. bulloides* shell geochemistry is also discussed.

**Key words.** *Globigerina bulloides*; Planktonic foraminifera; 16S rRNA gene metabarcoding; *Synechococcus*; symbiosis; endobiont; carbon isotopes



## 1. Introduction

Planktonic foraminifera are marine protists that precipitate an external shell composed of calcium carbonate. They are found at densities averaging 20–50 individuals m$^{-3}$ throughout the meso– and oligotrophic oceans, but can reach densities of >1,000 individuals m$^{-3}$ in polar ocean blooms (Schiebel and Hemleben, 2005; Kucera 2007; Lombard et al., 2009). Most planktonic foraminiferal species have rapid turnover rates with a generation time of a month or less (Hemleben et al., 1989; Bijma et al., 1990; Eretz et al., 1991; Schiebel et al., 1997; Lončarić et al., 2005; Jonkers et al., 2015) and are the source of up to 40 % of the biogenic carbonate exported from the surface ocean (Schiebel et al., 2002; 2007). Dissolution of this carbonate within the water column and sediments provides a significant buffering of ocean carbon chemistry and atmospheric CO$_2$ (Holligan and Robertson, 1996; Iglesias–Rodriguez et al., 2002; Schiebel, 2002; Freely et al., 2004; Ridgwell and Zeebe, 2005; Sarmiento and Gruber, 2006). Calcifying organisms like the planktonic foraminifera are under threat, however, from the continued release of anthropogenic CO$_2$ and associated changes in surface seawater pH due to ocean acidification (Ridgwell and Zeebe, 2005). Efforts have been made, therefore, to model the global abundance and distribution of the major species of planktonic foraminifera in an attempt to determine the impact of climate change on these organisms and to understand the implications for biogenic carbonate production over the coming decades (Lombard et al., 2011; Roy et al., 2015). However, the sensitivity of such models is hampered by the lack of basic ecological information (food preferences, symbiotic associations, response to changing calcite saturation states, mortality rates, etc.) required for their successful implementation (Roy et al., 2015).

The deposition and burial of planktonic foraminiferal calcitic shells at the sea floor generates a fossil record that dates back 180 million years (Hart et al., 2002). The geochemical signatures of their shells represent the contemporary physical and chemical nature of the water column in which they were precipitated. Their shell geochemical concentrations are used, therefore, as proxies for the reconstruction of past water column and climate conditions (Kucera 2007; Katz et al., 2010), allowing sensitivity testing of climate–models to refine climate change projections (Henderson, 2002). For accurate palaeoclimate reconstructions, it is important to understand the relationship between water column conditions and planktonic foraminiferal shell geochemical signatures. Individual species of extant planktonic foraminifera differ substantially in their environmental preferences and life histories (habitat, temperature, physiology, feeding, behaviour, reproduction, symbiotic assocations; e.g. Hemleben et al., 1989) which directly impacts on their shell geochemistry, resulting in the need for species-specific geochemical calibrations for some environmental factors, such as temperature (e.g. Erez and Luz 1983; Spero and Williams 1988; Spero et al., 1991; Bemis et al., 1998; 2000; 2002; Bijma et al., 1999; Anand et al., 2003; Eggins 2004; Russell et al., 2004). Many planktonic foraminifera also house protist algal symbionts (dinoflagellates or chrysophytes) within their cells (Bé et al., 1982; Spero, 1987; Gastrich, 1987; Hemleben et al., 1989; Lee and Anderson, 1991; Siano et al., 2010) that contribute photosynthetic products to the host (Gastrich and Bartha 1988; Caron et al., 1995; Uhle et al., 1997). Both symbiont photosynthesis and symbiont and host respiration alter the immediate chemical microenvironment, including



C and O isotope ratios ($\delta^{13}$C and $\delta^{18}$O) surrounding the host shell, which has been shown to influence shell geochemistry (Mashiotta et al., 1997, Rink et al., 1998; Wolf–Gladrow et al., 1999; Hönisch et al., 2003; Eggins et al., 2004).

Although the role and importance of the protist algal symbionts within planktonic foraminifera is widely recognized and relatively well understood, the association of planktonic foraminifera with bacteria has received very little scientific

attention. Apart from a few studies reporting the presence of living bacteria inside benthic foraminifera from dysoxic sediments (Bernhard et al., 2000; Bernhard et al., 2012; Tsuchiya et al., 2015), there has been little consideration of specific endosymbioses between foraminifera and prokaryotes. Indeed, there are no reports of planktonic foraminiferal relationships with bacteria other than a single report observing the external association of *Globigerinella siphonifera* Type I with the marine nitrogen–fixing, filamentous, cyanobacterium, *Trichodesmium* (Huber et al., 1997). This oversight is surprising, since

the occurrence of bacterial symbiosis within other protists is well established, as is their great potential for providing highly specialised metabolic processes to their hosts (e.g. Hoek et al., 2000; Schweikert and Meyer, 2001; Beier et al., 2002; Ashton et al., 2003; Foikin et al., 2003; Gast, 2009; Nowack et al., 2010; Orsi et al., 2012; Gilbert et al., 2012).

*G. bulloides* is a spinose planktonic foraminifer lacking protist algal symbionts (Febvre–Chevalier 1971; Spero and Lea, 1996) that is abundant in subpolar, temperate and low–latitude upwelling regions (Kleijne et al, 1989; Naidu and Malmgren,

1996). In these climatically sensitive areas it dominates the downward flux of foraminiferal shells to the sea floor and, as a consequence, is of considerable importance for palaeoclimate reconstructions (Sautter and Thunell 1991; Spero and Lea 1996). A complication in using *G. bulloides* for palaeoclimate reconstruction, however, is that despite its lack of protist algal symbionts, the shell geochemical signatures (e.g. $\delta^{18}$O and $\delta^{13}$C) of *G. bulloides* deviate from predicted values by more than any other extant, surface–dwelling species (Deuser et al, 1981; Kahn and Williams, 1981; Curry and Matthews, 1981; Kroon

and Darling, 1995; Spero et al, 1996; Bijma et al., 1999). Such large deviations are difficult to explain in the absence of protist algal symbionts, although they have been potentially linked to growth and ontogeny or even to *G. bulloides* respiration rates (Spero and Lea 1996). The presence of intracellular bacteria may provide an additional or contributing explanation.

The study of *G. bulloides* is further complicated by our current inability to distinguish morphologically the numerous

genotypes of *G. bulloides* identified within the morphospecies (Darling and Wade, 2008; Seears et al., 2012; Morard et al., 2013). The majority of the genotypes have been elevated to species level status (Andre et al., 2014) and all are potentially ecologically distinct, though they are commonly found in the same water column, where their adaptive ranges overlap (Darling and Wade, 2008; Morard et al., 2013). Where this occurs, aggregation of two ecologically distinct *G. bulloides* species could introduce significant noise into palaeoclimate calibrations (Darling et al., 2000; 2002), particularly if they

exhibit species–specific geochemical signatures.

In this study we have focussed on the cool water lineage *G. bulloides* Type IId, found throughout the year off the coast of California (Darling et al., 2003; Darling and Wade, 2008), which also corresponds to the region where the majority of experimental geochemical studies on the *G. bulloides* morphospecies have been carried out (e.g. Spero and Lea, 1996; Bemis et al., 1998; 2000; 2002). We examined the internal bacterial population of individual specimens of the planktonic



foraminifer *Globigerina bulloides* using a multiphasic approach of 18S rRNA gene sequencing, fluorescence microscopy, 16S rRNA gene metabarcoding via next–generation sequencing, transmission electron microscopy (TEM), and the polymerase chain reaction (PCR). We demonstrate that *G. bulloides* Type IId takes up picocyanobacterial endobionts (*Synechococcus* sp.) from the surrounding water column and enters into a species–specific association. We discuss the nature of this association and its potential metabolic and geochemical implications. We also discuss the power of the methodological approach taken for improving ecological knowledge of planktonic foraminifera.

## 2. Materials and Methods

### 2.1 Oceanographic setting

The Californian Current flows equatorward from the North Pacific Current (~50° N) to Baja California (~15–25° N). Southerly along–shore winds drive upwelling of cold nutrient–rich waters in early spring–summer in central California, and weaker but more sustained upwelling further toward the south. The relatively warm, saline Davidson Current and California Undercurrent flow poleward over the continental shelf. During the Southern California summer, the California Current moves farther offshore, and the Davidson Current predominates near shore (Checkley and Barth, 2009). For this study, samples were collected along the narrow Central California shelf ~1 km off Bodega Head, California (38.3° N, 123.0° W) and in the Southern California Bight off Santa Catalina Island (33.4° N, 118.4° W; Fig. 1). At both sites, local variation in foraminiferal abundances and species composition is well understood (Thunell and Sautter, 1992; Field, 2004, Davis et al., 2016), driven by periods of upwelling, relaxation or downwelling, and/or seasonal predominance of the Davidson Current. Further, the Santa Catalina Island site is close to the San Pedro Ocean Time Series (SPOT) station where the bacterial assemblage in the water column has been monitored both spatially and temporally for over a decade (Chow et al., 2013; Cram et al., 2015).

### 2.2 Sample collection

Samples were collected during July/August 2013, November 2014, and April 2015 (Table 1). The Bodega Head samples were obtained from vertically integrated 150µm mesh–size net tows, deployed to a maximum depth of 160 m, or to 10 m above the seafloor at shallower sites. Tow material was placed in ambient surface seawater and kept chilled during transit back to the Bodega Marine Laboratory where live foraminifera were wet picked. *G. bulloides* were then identified morphologically to the species level, rinsed in 0.6 µm filtered seawater and preserved in RNA*Later*® (Ambion™). This reagent conserves cell integrity, inhibits intracellular nucleases at ambient temperatures, and dissolves the calcite shell. The Santa Catalina Island samples were collected by scuba diving or net tows during July/Aug 2013. Collected foraminifera were treated as at the Bodega Marine Laboratory and transferred to RNA*Later*® at the Wrigley Marine Science Center.





## 2.3 Decalcification and washing of samples

To remove the shell and shell–associated, external contaminants, each individual specimen was decalcified by exposure to RNA*Later*® (Ambion[TM]). The cell was then washed in filter–sterilised, salt–adjusted phosphate buffered saline (PBS) or sterile artificial seawater, transferred to a new sterile 1.5 ml tube and washed a further three times before being transferred to DOC DNA extraction buffer (Sect. 2.4; Holzman and Pawlowski, 1996) for DNA analysis, or 4 % (w/v) paraformaldehyde in salt–adjusted PBS for microscopy.

## 2.4 Foram genotyping and Sanger DNA sequencing

DNA was extracted from individual foraminifer specimens using the DOC extraction method to identify the specific genotype (Holzman and Pawlowski, 1996). PCR was performed according to Seears et al., (2012). DNA sequencing was carried out using the BigDye® Terminator v3.1 Cycle Sequencing Kit and an ABI 3730 DNA sequencer (both Applied Biosystems).

## 2.5 DAPI staining and fluorescence microscopy

Foraminifer cells were stained with 4',6–diamadino–2–phenylindole (DAPI) which forms a highly fluorescent DAPI–DNA complex that allows the visualisation of bacterial cells and eukaryotic cell nuclei under fluorescence microscopy. Individual decalcified and washed foraminifer were fixed in 4 % (w/v) paraformaldehyde in salt–adjusted PBS for four hours at 4 °C. Fixed cells were transferred to a polylysine–coated microscope slide and dehydrated through an ethanol series of 70 %, 90 % and 100 % ethanol. Cells were stained in 1 µg ml[-1] DAPI (dilactate, Sigma–Aldrich) in PBS for three minutes and then rinsed with sterile deionised water. The stained preparations were mounted in AF1 mountant solution (Citifluor) and bacteria and eukaryotic nuclei visualised using a Zeiss Axio Imager Fluorescence microscope equipped with a DAPI filter set.

An unstained specimen of *G. bulloides* was also examined by fluorescence microscopy to observe the background levels of autofluorescence under the DAPI filter set to compare with the appearance of DAPI stained individuals. A TRITC filter set (excitation wavelength 550 nm, emission wavelength 620 nm) was used also on unstained individuals to investigate for the presence of autofluorescent, phycoerythrin–containing, cyanobacterial cells.

## 2.6 DNA extraction, amplification and 16S rRNA gene metabarcoding

DNA for 16S rRNA gene metabarcoding of the bacterial population within the foraminifera was extracted from decalcified and washed planktonic foraminiferal cells by the DOC extraction method (Holzman and Pawlowski, 1996). The V4 region of the 16S rRNA gene was amplified using the 515F forward primer and a barcoded 806R reverse primer series with the thermal cycling conditions detailed by Caporaso et al., (2012). PCR reactions contained 1 x *Taq* buffer plus additional MgCl$_2$ (final concentration 2.5 mM), 0.2 mM of each dNTP, 0.25 µM of each primer, 1 µl of template DNA and 1.25 U of *Taq* DNA polymerase (Roche Applied Science), with the volume made up to 25 µl with PCR grade water (Sigma). All PCR




reactions were set up in a PCR6 vertical Laminar Airflow Cabinet with UV sterilization (Labcaire Systems, Bristol, UK) as described by Pagaling et al., (2014). Reaction tubes and PCR mixtures were treated for 15 minutes with 15 W UV light (wavelength = 254 nm) to destroy contaminating DNA, prior to addition of dNTPs, *Taq* polymerase and template DNA (Padua et al., 1999). Three negative controls containing (i) no DNA template (two replicates) and (ii) DOC buffer only were

cycled alongside functional PCR reactions. The PCR reactions were run on a 1 % agarose gel and the products were purified with the Wizard® SV Gel and PCR Clean–Up System (Promega). The purified amplicons were quantified using a Quant–iT PicoGreen ds DNA Assay Kit (Life Technologies) prior to pooling samples at equimolar concentrations for DNA sequencing. The samples analysed were the PCR products from three independent *G. bulloides* isolates (BUL34, BUL36, BUL37) and a single non–spinose *Neogloboquadrina dutertrei* (DUT55) collected in July/Aug off Santa Catalina Island and

the products from two additional *G. bulloides* cells collected in November off Bodega Head (BUL22, BUL23; Table 1). The total number of quality–filtered sequencing reads including controls was 862,954. DNA sequencing was performed at Edinburgh Genomics using an Illumina MiSeq v3 to generate 250 base pair (bp) paired–end reads.

### 2.6.1 Quality filtering and contaminant removal

The Quantitative Insights in Microbial Ecology (QIIME, v1.8.0, Caporaso et al., 2010) pipeline was used to assemble

paired–end reads and quality filter the sequences. Raw reads were paired with an overlap of 200 bp and quality filtered with a minimum Phred score of 20 for maximum accuracy (Kozich et al., 2013). Reads of less than 245 bp (i.e. short reads) were removed from the dataset with the python script filter_short_reads.py from http://gist.github.com/walterst/7602058. Chimeras were detected using Usearch v6.1.544 default settings (Edgar at al., 2011) and version 13_8 of Greengenes 16S rRNA gene reference database (DeSantis et al., 2006). Given the low yield of endogenous bacterial DNA in these small–

sized samples, we anticipated that amplicon contamination from PCR amplification reagents, DNA extraction reagents, and the ultra–pure water system would contribute a significant number of DNA sequences and OTUs from contaminant genera to the sample set (Salter et al., 2014; Laurence et al., 2014). Operational taxonomic units with greater than 1000 sequences in any of the three control samples were considered to be potential contaminants and were removed from the sample set. Two OTUs were removed due to contamination in the two PCR controls; a *Bradyrhizobiaceae* of the class Alphaproteobacteria

and an *Acinetobacter* of the class Gammaproteobacteria. Twelve contaminating OTUs were removed due to contamination via the DOC buffer, seven of these were also of the class Alphaproteobacteria, order *Rhizobiales*: two additional *Bradyrhizobiaceae* OTUs; *Methylobacterium; Mesorhizobium; Pedomicrobium* and two further *Rhizobiales* OTUs; and one final OTU from the genus Alphaproteobacteria, *Sphingomonas* of the order *Sphingomonadales*. The final four OTUs were *Burkholdaria bryophila* of the class Betaproteobacteria*;* two *Sediminibacterium* OTUs of the phylum *Bacteriodetes;* and a

cyanobacterial OTU: *Streptophyta* chloroplast. A single *Bradyrhizobiaceae* OTU was by far the largest contaminant, with a total of 284,636 sequences from all samples and controls, and it is known to be a common contaminant of next–generation sequencing data, along with other Alphaproteobacteria (Laurence et al., 2014).



### 2.6.2 Operational taxonomic unit (OTU) picking and taxonomic assignment

The default QIIME pipeline was used for data analysis: OTU picking and taxonomic assignment. De novo picking (pick_de_novo_otus.py) clusters DNA sequences into OTUs with 97 % similarity with no external reference and selects a representative sequence of each OTU for alignment and subsequent assignment of taxonomy. This script keeps all diversity, including unknowns in the sample set. Closed reference picking was also performed which removes OTUs that are not closely matched (< 97 %) with OTUs in the Greengenes database (pick_closed_reference_otus.py). This output is required for normalisation by copy number (NBCN) using the online Galaxy tool ([http://huttenhower.sph.harvard.edu/galaxy/](http://huttenhower.sph.harvard.edu/galaxy/)). This corrects the abundance of each OTU to better reflect the true organism abundance by normalising predicted 16S rRNA gene copy number for each OTU. In both OTU picking methods OTUs with fewer than 10 sequences across all samples were removed from the sample set (filter_otus_from_otu_table.py).

### 2.6.3 Alpha–rarefaction and sequencing depth

In QIIME, the script alpha_rarefaction.py was used to assess whether the sequencing depth was adequate to detect foraminiferal bacterial diversity. Samples were rarefied to the lowest sequencing depth observed across all samples (10,551 in closed reference picking in sample BUL22) and OTU richness curves were generated, using the Observed Species metric which counts the number of unique OTUs found in a sample.

### 2.7 TEM

TEM was used to observe and document the structural relationships between the endobiotic bacteria and foraminiferal cells. Decalcified *G. bulloides* were fixed in 3 % glutaraldehyde in 0.1 M Sodium Cacodylate buffer, pH 7.3, for 2 hours followed by three 10 minute washes in 0.1 M Sodium Cacodylate. Specimens were then post–fixed in 1 % Osmium Tetroxide in 0.1 M Sodium Cacodylate for 45 minutes, followed by a further three 10 minute washes in 0.1 M Sodium Cacodylate buffer. Specimens were then dehydrated in 50 %, 70 %, 90 % and 100 % ethanol (X3) for 15 minutes each, then in two 10–minute changes in Propylene Oxide prior to being embedded in TAAB 812 resin. Sections, 1 μm thick were cut on a Leica Ultracut ultramicrotome, stained with Toluidine Blue, and then viewed under a light microscope to select suitable specimen areas for investigation. Ultrathin sections, 60 nm thick were cut from selected areas, stained in Uranyl Acetate and Lead Citrate and then viewed with a JEOL JEM–1400 Plus TEM.

### 2.8 Genetic identification of *Synechococcus* cells identified in *G. bulloides*

*Synechococcus* cells were found in large numbers inside *G. bulloides* and were genetically characterised. A 422 bp fragment of the *Synechococcus* 16S rRNA gene was amplified from total DNA extracted via the DOC method from individual specimen BUL34 (Table 1). This provided a larger, more informative fragment for phylogenetic analysis compared with the 253 bp generated by 16S rRNA metabarcoding. Cyanobacterial specific primers were used (CYA359f 5'–





GGGGAATCYTTCCGCAATGGG–3' and CYA781R a and b 5'–GACTACWGGGGTATCTAATCCCWTT–3', Nübel et al., 1997) and thermocycler conditions were as follows: 94 °C for 2 minutes followed by thirty cycles at 94 °C for 15 seconds, 55 °C for 15 seconds and 72 °C for 30 seconds followed by a final extension at 72 °C for 5 minutes. PCR reactions were performed with MyTaq REDDY mix (Bioline) and 0.25 µM of each primer, 1 µl of template DNA with the volume made up to 25 µl with PCR grade water (Sigma). The PCR product obtained was cloned (TOPO®–TA cloning kit, Invitrogen) and Sanger sequenced.

Clone sequences were aligned with reference *Synechococcus* 16S rRNA gene sequences retrieved from the GenBank database (NCBI) using ClustalW software within the package MEGA6 (Tamura et al., 2013). Phylogenetic trees (maximum likelihood; neighbour–joining; minimum evolution; UPGMA; maximum parsimony; Sect. 3.5) were generated using the default settings of MEGA6 with 500 bootstrap resamplings to determine the closest taxonomic affiliations (i.e., clade designation *sensu* Fuller et al., 2002) of the *G. bulloides*–associated *Synechococcus*. Informed by this analysis, further primers were designed that target other signature genes harboured by the *Synechococcus* clades identified including that for *rbcL*. This phylogenetically informative gene encodes the large subunit of RubisCO (ribulose–1,5–bisphosphate carboxylase/oxygenase), the primary $CO_2$–fixing enzyme found in cyanobacteria. Primers (SynrbcL_For 5'– CGGCAACTTCTTCGATCAGG–3'; SynrbcL_Rev1 5'– ATGTCGCGGCTTTCTTTCTC–3'; SynrbcL_Rev2 5'– CCGGCTTCCATAAGGATGTC–3') were designed with Primer 3 (http://primer3.ut.ee/) that target a 252 bp fragment of *rbcL* from the most closely related *Synechococcus* spp.(i.e., strains CC9902, CC9311 and WH8120, see below).

Purified DNA from fourteen *G. bulloides* specimens (Table 1) generated products of the correct size on PCR amplification with the *rbcL* primers. The product obtained from isolate BUL34 was selected and TA cloned and DNA sequenced as described above. PCR reactions were performed in a Biometra Personal Thermocycler using MyTaq REDDY mix (Bioline) and 0.25 µM of each primer, 1 µl of template DNA with the volume made up to 25 µl with PCR grade water (Sigma). Thermocycling conditions were 94 °C for 2 minutes followed by 30 cycles at 94 °C for 15 seconds, 56 °C for 15 seconds and 72 °C for 30 seconds followed by a final extension at 72 °C for 5 minutes.

## 3. Results

In total, 29 individual specimens of *G. bulloides* collected from waters off Santa Catalina Island and Bodega Head (Fig. 1) were investigated during this study. The sampling information and analyses performed on each specimen is detailed in Table 1, and the sampling strategy and genetic characterisation are described in the methods.

### 3.1 Genotyping of foraminifera

Partial 18S rRNA (SSU) gene sequences amplified from BUL34 and DUT55 have been submitted to Genbank (NCBI, accession numbers KX816046 and KX816048 respectively). *G. bulloides* specimen BUL34 is Type IId and *N. dutertrei*





specimen DUT55 is Type Ic and it is the first time a ~1000 bp fragment has been amplified for this genotype. Both genotypes have been found routinely in the Southern California Bight (Darling et al., 2003).

## 3.2 Fluorescence microscopy and DAPI staining

Fluorescence microscopy examination of an unstained, fixed *G. bulloides* specimen (BUL21; Table 1) under the DAPI filter set demonstrated high levels of diffuse autofluorescence across the entire cell. *G. bulloides* cells that were first stained with DAPI (n=6; Table 1) also showed background autofluorescence but in addition some more highly fluorescent regions 3−10 μm in size were also observed (Fig. 2a). These stained structures are probably DNA from the foraminiferal nucleus and also DAPI–DNA complexes in organisms sequestered within food vacuoles. Of greater present significance, however, were the very many fluorescent, globular structures of approximately 1 μm diameter observed consistently within all of the *G. bulloides* cells analysed. Their small, regular size is consistent with the presence of intact intracellular (coccoid) bacteria residing within the foraminiferal cell (Fig. 2b). Further microscopic examination of unstained *G. bulloides* using the TRITC filter set also revealed high autofluorescence across the cell but, in addition, brighter fluorescence from many of these approximately 1 μm diameter DNA–containing structures. This observation is entirely consistent with the presence of the photosynthetic pigment, phycoerythrin (excitation maxima ~495, 545 nm, emission maximum 565−580 nm), within many of these intracellular (cyano)bacteria (see below).

## 3.3 16S rRNA gene metabarcoding

16S rRNA gene metabarcoding was carried out on five specimens of *G. bulloides* and a single specimen of the non–spinose species *Neogloboquadrina dutertrei* for comparison (Table 1). The raw dataset has been submitted to the sequencing read archive (SRA, NCBI), Bioproject accession number SRP090165, SRA accession numbers: SRR4271458, SRR4271479, SRR4271493, SRR4271505, SRR4271506, SRR4271507). A total of 862,954 sequences was generated from the six samples and three controls after quality filtering, and removal of short reads (<245 bp) and chimeric sequences (Sect. 2.6.1). In closed reference picking, after removal of pynast failures, all control sequences (288,985) contaminant OTUs (including 161,282 sequences as result of the single contaminating *Bradyrhizobiaceae* OTU across all six samples) and OTUs with an abundance of less than 10 sequences across all samples, a total of 214,087 sequences were clustered into OTUs and assigned taxonomy (Sect. 2.6.1 and 2.6.2). The numbers of sequences and OTUs generated in individual specimens for both closed reference picking and de novo picking are listed in supplementary Table S1. The OTU profiles within a specimen were highly similar between de novo picking and closed reference picking with normalisation by copy number (NBCN, Sect. 2.6.2). Therefore, we present results for closed reference picking with NBCN, and indicate when de novo picking OTUs are being represented. Rarefaction curves (supplementary Fig. S1) for OTU richness confirmed that sequencing depth was sufficient to capture the full bacterial assemblage diversity.

All replicates of *G. bulloides* contained a highly distinctive assemblage of OTUs. The OTU assemblage of an individual *N. dutertrei* (Table 1) sampled at the same time is shown for comparison (Fig. 3). Within the five BUL specimens investigated





(Table 1), 37–87 % of all sequences belonged to five OTUs assigned to the unicellular, cyanobacterial genus, *Synechococcus* (Fig. 3). This was by far the greatest abundance of a single genus of bacteria; not another bacterial genus, or, indeed, family or order was found across all five BUL specimens at relative abundances consistently more than 2 %. The next highest relative abundance group, therefore, must be described at the class level. Across four (BUL22, BUL23, BUL34, BUL36) of

the five BUL specimens 15–31 % of sequences belonged to the class Alphaproteobacteria and were dispersed amongst 81 OTUs. The fifth and outlying specimen (BUL37, containing 87 % *Synechococcus* sequences)*,* contained only 2.6 % Alphaproteobacteria across 37 OTUs. It contained marginally more phylum Actinobacteria (1.8 %) and class Betaproteobacteria (1.4 %) sequences, but the sequence abundances of these classes were more similar to the other BUL specimens. The high relative abundance of *Synechococcus* within this specimen might be due to a lack of feeding on other

bacteria or algae immediately prior to sampling, indicative of a high turnover rate for prey bacteria and algal cells.

There were no chloroplast–affiliated OTUs in specimens BUL22 and BUL23. However, 6.4 %, 27 % and 3 % of sequences in specimens BUL34, BUL36 and BUL37 respectively were allocated to chloroplast 16S rRNA gene OTUs from a variety of sources. These were two OTUs from a mixotrophic protist belonging to the diverse, protozoan phylum, Cercozoa (Cavalier–Smith and Chao 2003); three OTUs from the phylum of Haptophyte algae that includes coccolithophores; eight OTUs from

the phylum Stramenopile that includes both diatoms and chrysophyte algae; and, finally 13 OTUs from the group Streptophyta (an unranked clade of plants that includes green algae).

In de novo picking, *G. bulloides* specimens BUL34, BUL36 and BUL37 contained varying percentages of sequences (25.4 %, 0.3 % and 1.6 % and respectively) in three taxonomically unassigned OTUs whereas BUL22 and BUL23 did not contain any unassigned sequences. 94 % of all the unassigned sequences belonged to a single OTU (e.g. 5,878 sequences in sample

BUL34) that was 99 % identical (100 % coverage of 253 bp) to an unidentified marine bacterial clone (accession number HQ673258) retrieved from the northeast subarctic Pacific Ocean (Allers et al., 2013). The nearest match (87–89 % similarity over 100 % coverage) to an identified phylum was to a large number of uncultured Verrucomicrobiae bacteria of the phylum Verrucomicrobia from a wide range of habitats, including marine environments. Whilst this was not a particularly close match, it is within the defined limit of >85 % DNA identity that delineates a phylum (Hugenholtz et al., 1998; Rappé and

Giovanonni, 2003) albeit based only on a 253 bp fragment. The variation in abundances of this OTU found in each *G. bulloides* individual analysed (25 % in BUL34; 0.3 % in BUL36 and 1.6 % in BUL37) might indicate this bacterium has a patchy distribution and is an opportunistic food source.

### 3.4 Transmission electron microscopy

Transmission electron microscopy (TEM) imaging was carried out on four *G. bulloides* specimens (BUL32, BUL39,

BUL69, BUL71; Table 1, Fig. 4) to observe whether any endobionts were present within the cell. No intracellular eukaryotic cells were observed, confirming a lack of algal symbionts. However, numerous intact coccoid cells containing carboxysomes (Fig. 4a) and surrounded by thylakoid membranes, characteristic of the cyanobacterium, *Synechococcus*, were observed





throughout the cytoplasm and also in vacuoles of all individual *G. bulloides* observed (Fig. 4b). Approximately 5 % of the observed *Synechococcus* cells were undergoing cell division (Fig. 4c).

To compare foraminiferal cellular *Synechococcus* concentrations with those of the water column, the concentration of *Synechococcus* cells ml$^{-1}$ of foraminiferal cytoplasm was calculated by assuming a conservative average host cell diameter of

200 µm (Spero and Lea, 1996; Aldridge et al., 2012), a spheroid morphology (Geslin et al., 2011) and that the cytoplasm was equivalent to 75 % of the shell volume (Hannah et al., 1994). Based on averaged cell counts from the TEM images, the total number of *Synechococcus* cells within *G. bulloides* occupied less than 2 % of the foraminiferal cell volume but was equivalent to 3.8 x10$^9$ *Synechococcus* cells ml$^{-1}$, compared with between $1 \times 10^2 - 1 \times 10^6$ *Synechococcus* cells ml$^{-1}$ of seawater throughout the global ocean (Partensky et al., 1999; Paerl et al., 2011). In the Southern California Bight, *Synechococcus* cell

counts are generally fewer than $1 \times 10^5$ cells ml$^{-1}$ but can reach $2.5 \times 10^5$ cells ml$^{-1}$ during blooms (Tai and Palenik 2009; Tai et al., 2011). The concentration of *Synechococcus* in the *G. bulloides* cell, therefore, was up to 4 orders of magnitude greater than that in the surrounding water column, indicating that *G. bulloides* selectively concentrate *Synechococcus* in their cytoplasm.

### 3.5 Genetic characterisation of intracellular *Synechococcus*

Five *Synechococcus* OTUs were assigned in 16S rRNA gene metabarcoding with closed reference picking. However, more than 99 % of the BUL *Synechococcus* sequences were assigned to just one of these OTUs. The representative nucleotide sequence (253 bp) of this OTU is a 100 % match to the coastal, clade IV *Synechococcus* sp. strain CC9902. Two further OTUs were highly similar to this abundant OTU and were 99% identical to *Synechococcus* sp. strain CC9902. The remaining two OTUs both had a nucleotide match of 99 % with *Synechococcus* sp. Strain WH8020, a Clade I strain also

found typically in coastal waters. In order to confirm these clade assignments, phylogenetic analysis (supplementary Fig. S2) of a larger (425 bp) fragment of the *Synechococcus* 16S rRNA gene generated from BUL34 total DNA was performed. Ten clones (GenBank accession numbers KX815969–KX815978) clustered with the Clade IV *Synechococcus* strain CC9902 and two clones (GenBank accession numbers KX815979 and KX815980) clustered with Clade I strains CC9311 and WH8120, in agreement with the 16S rRNA gene metabarcoding data. The topologies of the phylogenetic trees produced were all in

overall agreement with well–established analyses of *Synechococcus* 16S rRNA genes (Scanlan et al., 2009) confirming the phylogenetic resolution of the sequence data included in the present study.

In addition, a 252 bp fragment of the *Synechococcus* rbcL gene was cloned and 230 bp of this clone was DNA sequenced (GenBank accession number KX816048) from BUL34 (Table 1). A GenBank BLAST search (NCBI) found 100 % nucleotide sequence identity with the RuBisCo large subunit coding region of *Synechococcus* CC9902 and 92 % identity

with *Synechococcus* WH8020, confirming the presence of *Synechococcus* CC9902, or a very closely related Clade IV strain. The DNA of thirteen further *G. bulloides* specimens (Table 1) also yielded products of ~252 bp on amplification with the *Synechococcus* rbcL primers confirming the consistency of the association between *G. bulloides* Type IId and *Synechococcus* strains in the California Current year round.



## 4. Discussion

Our results highlight a novel endobiotic association between the usually free–living, photoautotrophic picocyanobacterium, *Synechococcus* spp., and its host, *G. bulloides* Type IId, a genotype of a spinose planktonic foraminiferal morphospecies, barren of protist algal symbionts. Below, we discuss the evidence for this endobiosis, and possible roles of *Synechococcus* in *G. bulloides* host metabolism and its characteristic cytoplasm colouration. A better understanding of *G. bulloides* genotype ecology will ultimately provide ecological information for modelling foraminiferal distribution, abundance and seasonality under different climate regimes, and improve the accuracy of the palaeoceanographic proxy records.

### 4.1 Evidence for *Synechococcus* as an abundant endobiont of *Globigerina bulloides* Type IId

*Globigerina bulloides* has consistently been reported to be barren of protist algal symbionts (Febvre–Chevalier 1971; Gastrich 1987; Hemleben et al, 1989; Spero and Lea, 1996). The current study supports this conclusion, since no intact algal cells were found in any of the *G. bulloides* cell sections examined using TEM. However, we do have strong evidence that *G. bulloides* Type IId contains large numbers of the photoautotrophic picocyanobacterium, *Synechococcus,* and that they are preferentially taken up from the water column by *G. bulloides* and concentrated within the host cytoplasm. Based on the observations discussed below, we propose that these picocyanobacteria are abundant, metabolically active endobionts living within the *G. bulloides* cell, rather than prey.

### 4.1.1 *Synechococcus* cells are intact and viable

TEM images showed that the *Synechococcus* cell membranes appeared to be physically intact (Fig. 4a) and, whilst some *Synechococcus* cells were observed within vacuoles, many were distributed throughout the cytoplasm of *G. bulloides* (Fig. 4b) where digestion does not occur. As many as 5 % of the intracellular *Synechococcus* population were observed to be in the process of cell division (Fig. 4c) indicative of actively growing, viable individuals (Campbell and Carpenter, 1986). Significantly, Bernhard et al., (2000) considered as few as 3 % dividing cells a substantial enough proportion to suggest a symbiotic role for the intracellular bacteria they observed in the benthic foraminifer *Buliminella tenuata*. Further, autofluorescence in the orange/red spectral region arising from the photosynthetic pigment phycoerythrin, was readily detected within these DNA–containing endobionts within *G. bulloides*. Phycoerythrin, a water–soluble biliprotein found routinely in marine *Synechococcus* spp. rapidly diffuses into the surrounding aqueous milieu if the cell membranes are compromised (Stewart and Farmer, 1984; Wyman, 1992). In further confirmation of the intact nature of the intracellular population, two partial *Synechococcus* genes (the 16S rRNA gene and *rbcL*) were successfully amplified by the PCR, providing additional evidence that *Synechococcus* DNA was not grossly degraded by nucleases.





### 4.1.2 *Synechococcus* are endobionts in marine protists

Whilst *Synechococcus* are known primarily as free–living organisms (Waterbury et al, 1979; Richardson and Jackson 2007), an endobiotic lifestyle has also been observed in association with a number of different marine protist groups. *Synechococcus* has been identified in the benthic foraminifer *Fursenkoina rotundata*, sampled from the benthos at 600 m

using both fluorescence microscopy (Bernhard et al., 2000) and TEM imaging (Seckbach, 2006). At these depths, however, the *Synechococcus* endobionts would be unable to photosynthesise, which rules out the most obvious functional role for this potential symbiont. *Synechococcus* have also been found living embedded within the extracellular matrix surrounding a marine diatom (Buck and Bentham 1998), and within a polycystine radiolarian (Yuasa et al., 2012). This study now confirms that they are also to be found within the living cells of at least one type of planktonic foraminifer.

### 4.1.3 *Synechococcus* cells are specifically taken up from the water column and accumulate in the *G. bulloides* cytoplasm

Not only are the intracellular *Synechococcus* cells intact, but they are also selectively accumulated within the *G. bulloides* cytoplasm at densities (~3.8x10$^9$ cells ml$^{-1}$) that are four orders of magnitude more concentrated than reported in the surrounding seawater (Tai and Palenik 2009; Tai et al, 2011). Whilst DNA sequences from other bacteria were identified by

16S rRNA gene metabarcoding (Fig. 3), no bacterial cells lacking carboxysomes were observed by TEM, indicating that, unlike *Synechococcus*, other bacteria were rapidly digested once taken up. Quite how *G. bulloides* first selects and then accumulates *Synechococcus* cells, however, is uncertain. In the case of planktonic foraminifera harbouring protist algae, the symbionts are taken up directly from the water column (horizontal transmission) rather than being inherited through vertical transmission via parental gametes (Hemleben et al, 1989; Bijma et al, 1990). Juveniles with only 2 to 3 chambers already

have ~3 to 5 symbionts, and it is assumed that they are taken up from the water column exclusively since no protist symbionts (5–10 μm cell diameter) have been observed within the much smaller flagellated gametes (~2.5 μm; Hemleben et al, 1989). Although picocyanobacteria such as *Synechococcus* are much smaller in size (~1 μm) than algal symbionts and could potentially be inherited via parental gametes, we favour the hypothesis that the *Synechococcus* population within *G. bulloides* is similarly taken up from the water column, despite evidence for both horizontal (Ashton et al., 2003) and vertical

transmission (Schweikert and Meyer, 2001) of bacteria within protist hosts.

To investigate the potential mode of transmission of the *Synechococcus*, we compared the strain assemblages within *G. bulloides* with those of the surrounding water column. If the *Synechococcus* endobionts were horizontally transferred to *G. bulloides* via selective uptake from the water column, we would expect that the diversity of the internal strain assemblage would mirror that of the surrounding waters. Alternatively, if the endobionts were vertically transmitted, a degree of genetic

drift would be expected between the internal and free–living strains of *Synechococcus* as the result of genetic isolation over time (Wernegreen, 2002; Bright and Bulgheresi, 2010). Off the coast of California, the most prevalent strains of *Synechococcus* are those belonging to Clades I and IV (see Fuller et al., 2003) that display seasonal population differences throughout the annual cycle (Tai and Palenik, 2009). The *Synechococcus* 16S rRNA gene sequences cloned from a *G.*



*bulloides* specimen collected in July/Aug (Table 1) show that the strain composition strongly reflects the seasonal cladal distribution patterns that are observed in the water column at that time of year (Tai and Palenik, 2009). Up to 100 % nucleotide identity was found for the 16S rRNA gene clones and the *rbcL* gene sequences of the internal endobionts and those of the free–living, clade IV *Synechococcus* strain CC9902, originally isolated from waters off the California coast. This strongly supports a strategy of horizontal rather than vertical transmission for the *G. bulloides* endobionts.

### 4.1.4 Intracellular OTU relative abundances do not reflect those of the water column

The intracellular 16S rRNA gene OTU profiles of *G. bulloides* were very different from those of the water column assemblages, indicating very specific uptake of bacteria from the general microbial population. The foraminifera collection site off Santa Catalina Island in the San Pedro Channel is adjacent to the SPOT sampling location (33° 33'N, 118° 24'W), where seasonality and trophic interactions within the microbial assemblages in the water column have been studied routinely for over a decade (Chow et al., 2013; Cram et al., 2015). In both the surface waters and deep chlorophyll maximum layer, the microbial assemblage at the SPOT sampling site is dominated by OTUs of the ubiquitous SAR11 group (Giovannoni 1990; Morris et al., 2002) of marine Alphaproteobacteria, that represent over 30 % of the assemblage. In addition, members of the Actinobacteria account for approximately 15 % of OTUs while the picocyanobacteria represent just 2−5 % of the total bacterioplankton. Of the latter, *Prochlorococcus* dominates the assemblage although *Synechococcus* is also present year round (Chow et al., 2013). The remaining 50 % of the microbial population comprises a series of OTUs from a variety of marine bacteria each representing less than 2 % of the assemblage (Chow et al., 2013). This water column assemblage contrasts strongly with the intracellular 16S rRNA gene OTUs of *G. bulloides*, where between 37 % and 87 % of the total number of sequences recovered belong to *Synechococcus* OTUs. Strikingly, *Prochlorococcus* sequences were not identified in the three *G. bulloides* specimens collected close to the SPOT sampling location (BUL34, BUL36 and BUL37), even though *Prochlorococcus* represents the majority of the picocyanobacteria in the water column in this region. Further, < 4.5 % of OTUs in the amplified *G. bulloides* specimens were assigned to the Actinobacteria (compared to ~15 % in the water column) and no OTUs of the ubiquitous SAR11 group of Alphaproteobacteria were identified in our sample set. However, this is likely to be a result of bias against SAR11 clades (Apprill et al., 2015; Walters et al., 2015) in the primer set used in this study (Caporaso et al., 2012).

The composition of the internal microbial population of the *G. bulloides* cells clearly does not mirror that of the surrounding water column, highlighting the species–specific nature of the OTU assemblages observed within the *G. bulloides* cell. This observation is also reinforced by the fact that the intracellular OTUs within *G. bulloides* also differ substantially from those identified within specimens of the non–spinose species *N. dutertrei* (for e.g. DUT55; Fig. 3), collected at the same time and location. *N. dutertrei* contains ~ 2 % bacterial OTUs with the majority of OTUs (> 97 % being assigned to Stramenopiles (a group which includes diatoms and chrysophyte algae; 53 %) and Cercozoa (a diverse phylum of mixotrophic protists; 44.5 %). This highlights again the species–specific nature of the *G. bulloides* intracellular OTU assemblage.





### 4.1.5 Unusual cytoplasm colouration of *G. bulloides*: a role for endobiotic *Synechococcus*

In the specimens found off the coast of California, living *G. bulloides* cells often exhibit a brown colouration (Spero and Lea, 1996), a colour that does not generally characterise the other spinose species in the region. The discovery of phycoerythrin–containing *Synechococcus* spp. within the cytoplasm of the foraminifera reported here provides a plausible explanation for this unusual colouration. The clade I and IV *Synechococcus* strains with which the *G. bulloides* endobionts cluster in phylogenetic analysis, are all so–called type IV chromatic adapters (Six et al., 2007). These organisms have a distinct yellow/orange appearance when grown in blue light; (i.e., under the illumination conditions typical of the oligotrophic waters off the California coast from which the samples were obtained during the present study) owing to the production of urobilin–rich phycoerythrins under this light regime and also to elevated concentrations of the photoprotective carotenoid, zeaxanthin, (Bidigare et al., 1989). The presence of these pigments, therefore, probably contributes to the unusual cytoplasm colouration of the host.

### 4.2 Potential metabolic roles for the *G. bulloides* endobionts

There are some obvious potential metabolic benefits to each organism in the *G. bulloides–Synechococcus* partnership. Firstly, the foraminifer might benefit from a supply of photosynthetically fixed carbon, as is the case with the foraminifera that harbour protist algal symbionts (Caron et al., 1995; Uhle et al., 1997; 1999). If this were the sole benefit, however, one would question why *G. bulloides* preferentially recruits *Synechococcus* for this purpose, rather than the more conventional algal symbionts found in other species. One possible explanation is that *G. bulloides* inhabits a wide range of depths that often extend below the photic zone and it is also common in unstable upwelling waters where potential algal symbionts may not thrive. *Synechococcus* has been found alive in aphotic waters at depths of 600m (Bernhard et al., 2000), and it is capable of assimilating carbon mixotrophically (Paoli et al, 2008). It could therefore switch between phototrophy and (photo)heterotrophy, depending on the water column depth of the host. In addition, the clade I and IV groups of *Synechococcus,* as found in *G. bulloides*, are chromatic adapters, able to modify their pigment composition and absorption properties depending on the underwater light field (Six et al., 2007). Such adaptability might make *Synechococcus* a more compatible symbiont for the *G. bulloides* lifestyle.

Alternatively, *Synechococcus* may have additional or quite separate functional roles in association with *G. bulloides* beside endobiotic photosynthetic activity within the photic zone. For example, approximately half of the nitrogen assimilated by the host cell in the *Orbulina universa* foraminifer–symbiont system is transferred via the algal symbionts; a contribution that increases further to ~90–100 % in nitrate–depleted waters (Uhle et al., 1999). *Synechococcus* spp. have a very high affinity for combined nitrogen (for e.g. nitrate, nitrite and ammonium) and accumulate expanded stores of this element within their light–harvesting phycobilisomes under N–replete conditions (Wyman et al., 1985). Likewise, *Synechococcus* sequester large stores of P within their cells as polyphosphate, even under low external concentrations (Martin et al., 2014). These nutrient reservoirs could be readily mobilised and exploited by the foraminiferal cell, particularly prior to gametogenesis, when



planktonic foraminifera require extra elemental resources for DNA production (Hemleben et al., 1989). For *Synechococcus*, being housed within a foraminiferal cell could protect it from grazers and the multitude of cyanophages present in the water column (Suttle and Chan 1994; Mühling et al, 2005). *Synechococcus* may also benefit presumably from a supply of host metabolic by–products or from specific nutrients as products of prey digestion.

## 4.3 Feeding preferences and life strategy of *G. bulloides* Type IId

TEM in combination with 16S rRNA gene metabarcoding enables identification of both bacteria and eukaryotic chloroplasts within the foraminiferal cell. This method does not amplify eukaryotic, nuclear–encoded, (18S) rRNA genes and, as a result, does not provide any information about the non–chloroplast–bearing zooplankton prey of *G. bulloides*. Observations of large numbers of freshly collected specimens of *G. bulloides* confirm that they feed on small zooplankton prey as well as phytoplankton (Spero and Lea, 1996). Amongst the latter, they have a preference for some species of diatoms and chlorophytes over dinoflagellates or chrysophytes (Lee et al., 1966). Interestingly, however, two of the five *G. bulloides* specimens in this metabarcoding study (BUL22, BUL23) did not contain any chloroplast DNA, indicating that they had not fed on phytoplankton prior to sampling. However, these specimens were sampled in November (Off Bodega Head) from vertically integrated net tows and may have been obtained from resident populations as deep as 150 m, while the three in which chloroplast 16S rRNA sequences were present (BUL34, BUL36 and BUL37) were sampled from shallow water nets in July/Aug (off Santa Catalina Island). These differences in OTU composition, therefore, could be as a result of location, depth or seasonal differences in available diet. The three *G. bulloides* with chloroplast sequences (6.4 %, 27 % and 3 %, respectively) were clearly feeding on a range of photosynthesising eukaryotes (Sect. 3.3). OTUs indicate these to be Cercozoa (mixotrophic protists), Streptophyta (includes green algae), Haptophyta (includes coccolithophores) and Stramenopiles (includes both diatoms and chrysophyte algae).

Our data suggest that *G. bulloides* may also utilise bacteria as a significant food source. *G. bulloides* contained 33.7–62.5 % of non–*Synechococcus* bacterial sequences within the cell, (BUL37 was an outlier with only 10 %, Fig. 3) corresponding to a diverse assemblage of 200 OTUs. We assume that these sequences are derived from prey species because no intact bacteria lacking the carboxysomes and thylakoid membranes found in Synechococcus were observed in TEM images of the *G. bulloides* cytoplasm or non–digestive vacuoles. The most abundant group of sequences recovered (15–31 %, outlier BUL37 contained 2.6 %) comprise 81 OTUs belonging to the class Alphaproteobacteria, perhaps indicating a preferential selection of specific members within this class. The remaining 17–47.5 % (outlier BUL37, 7.5 %) of sequences were made up of a diverse collection representing other major phyla of bacteria (Sect. 3.3; Fig. 3) that include the Acidobacteria, Actinobacteria, Bacteriodetes, Firmicutes and Planctomycetes and the classes Beta–and Gammaproteobacteria of the phylum Proteobacteria.

*G. bulloides* Type IId is found throughout the year in the Southern California Bight, where it is exposed to cool, upwelling periods of high productivity and also to warmer periods characterised by more stratified, less productive conditions (Darling et al., 2003; Darling and Wade, 2008). It has a relatively high growth rate, possibly reproducing within 2–3 weeks (Spero





and Lea, 1996; Lombard et al., 2009). In combination with our data, this suggests that *G. bulloides* Type IId is a generalist predator with an opportunistic feeding strategy, utilizing bacterioplankton as well as phyto– and zooplankton, the proportions of which may be seasonal and depth dependent. Such opportunism may enable *G. bulloides* to grow and reproduce rapidly within its diverse habitat. We propose two hypotheses for the life strategy of *G. bulloides* Type IId to

survive the challenges presented within the broad seasonal changes in the region. The first is that it is a mixotrophic feeder (Mitra et al., 2016) and that the *Synechococcus* endobionts are photosynthesising symbionts contributing fixed carbon to the foraminiferal host. In this scenario, they would be fulfilling a functional role similar to that of algal symbionts in other spinose species. Alternatively, or concurrently, *Synechococcus* may be exploited by *G. bulloides* Type IId for its nutrient assimilation and storage capacity and then digested as an extra energy, nitrogen and phosphate source for DNA replication at

reproduction.

### 4.4 The importance of genotype ecology

Since *G. bulloides* occurs in great abundance in cool, high latitudes and at mid to lower latitude upwelling systems (Kleijne et al., 1989; Naidu and Malmgren, 1996), it is one of the most commonly used planktonic foraminifera for palaeoclimate reconstruction (Sautter and Thunell 1991; Spero and Lea 1996). In order to reconstruct past changes in oceanic conditions

using the shell geochemical data, it is important to obtain a thorough understanding of the relationship between foraminiferal ecology and the geochemistry of its shell. This relationship is based on the assumption that each foraminiferal morphospecies represents a genetically continuous species with a unique habitat preference. However, since *G. bulloides* inhabits such a wide range of different ecosystems, it is not surprising that several ecologically distinct genotypes have been recognised (Darling et al., 1999; Kucera and Darling 2002; Darling and Wade 2008; Seears et al., 2012; Morard et al., 2013).

Indeed, recent species delineation studies support species status for several of the *G. bulloides* genotypes (André et al., 2014) including *G. bulloides* Type IId. Such diversity could result in genotype–specific geochemical signatures across the morphospecies (Healy–Williams 1985; Bijma et al., 1998; de Vargas 2001; Kucera and Kennet, 2002). Both Kucera and Darling, (2002) and Morard et al., (2013) have demonstrated that, based on ecological knowledge, integrating *G. bulloides* genotypes into assemblage–based SST reconstructions significantly improves resolution. This demonstrates the value in

understanding the ecology of genotypes within a morphospecies and the necessity of establishing whether the association between *G. bulloides* Type IId and *Synechococcus* is universal across this foraminiferal morphospecies complex.

### 4.5 Implications for palaeoceanography

The discovery of intracellular bacteria within a palaeoceanographically significant foraminiferal host may lead to a significant improvement in our current understanding of foraminifer shell geochemistry. The carbon isotopic composition of

planktonic foraminifera has the potential to help reconstruct changes in the chemocline of the surface ocean, providing insights into changes in ocean circulation (Spero et al., 2003). However, the interpretation of $\delta^{13}C$ data is often inhibited by poor understanding of the causes of offsets between shell $\delta^{13}C$ and the $\delta^{13}C$ of the dissolved inorganic carbon from which



foraminifera build their shells. In particular, the $\delta^{13}C$ of *G. bulloides* shells deviates from predicted values more than that of any other extant species (Deuser et al., 1981; Kahn and Williams, 1981; Curry and Matthews, 1981; Kroon and Darling, 1995; Spero and Lea, 1996; Bijma et al., 1999), implying consistent use of metabolic carbon during calcification by this morphospecies (Deuser et al., 1981).

Symbiont photosynthesis as well as symbiont and host respiration alter the chemical microenvironment surrounding the host shell, which in turn influences their shell geochemical signatures (Rink et al., 1998; Wolf–Gladrow et al., 1999; Eggins et al., 2004). In this *G. bulloides*/*Synechococcus* association, respiration of both endobiont and host would contribute $^{13}C$–depleted $CO_2$ to the calcifying microenvironment (Spero and Lea, 1996), whilst *Synechococcus* photosynthesis would counteract this by preferentially removing $^{12}CO_2$ and hence elevating $^{13}C/^{12}C$ ratios in the remaining dissolved $CO_2$, as occurs in protist algal

symbiont bearing planktonic foraminifera (Spero et al., 1997). The large offset towards $^{13}C$–depleted values measured in *G. bulloides* suggests that *Synechococcus* respiration dominates the shell geochemical signature, and implies that photosynthesis is not the primary role of *Synechococcus* in this association (Sect. 4.2). The presence of metabolically active *Synechococcus* in *G. bulloides* Type IId, therefore, may account for the unusual shell $\delta^{13}C$ determined via culture–based studies conducted at Santa Catalina Island. Since *G. bulloides* Type IId is abundant here, it is unlikely that variation in genotype has contributed

to uncertainties in these calibrations, but applying these culture–based calibrations to other regions in which different genotypes dominate may produce erroneous results (Darling et al., 2003). It is of particular importance therefore, to determine whether the *Synechococcus/G. bulloides* association exists in other *G. bulloides* genotypes in order to generate and apply genotype–specific palaeoclimate calibrations.

## 5. Conclusions

This is the first report of bacterial endobionts within a planktonic foraminiferal species. Our results show that the picocyanobacteria *Synechococcus* spp. are found in large numbers within the protist algal symbiont–barren foraminifer, *G. bulloides*. *Synechococcus* spp. are taken up from the water column by the host and live and divide within the host cytoplasm at substantially higher concentrations (~4 orders of magnitude) than those found in the surrounding seawater. Their role is not yet known, but their potential for both phototrophy and (photo)heterotrophy makes *Synechococcus* an ideal symbiont for

*G. bulloides* as it occupies water depths both within and below the photic zone. Additionally, the ability of *Synechococcus* to store P as polyphosphate, and N within biliproteins under nitrogen replete conditions, would be beneficial for a foraminiferal host exhibiting fast reproductive turnover, with a high nutrient and energy demand at gametogenesis. Further experiments are required on the *G. bulloides* Type IId/*Synechococcus* association to elucidate the full relationship between the two organisms. More investigations are also needed of the *G. bulloides* morphospecies globally, to determine how widespread

the association is to improve understanding and accuracy of this species as a palaeoclimate proxy.

In this study we have demonstrated that 16S rRNA gene metabarcoding of the intracellular DNA of planktonic foraminifera and TEM has the potential to provide new insights into the biological associations and seasonal feeding preferences of

ecologically distinct genotypes of planktonic foraminifera. With the addition of 18S rRNA gene metabarcoding to target protist and multicellular zooplankton, next generation DNA sequencing technologies could transform the usefulness and accuracy of planktonic foraminiferal global distribution and seasonality models by providing the essential ecological information currently unavailable (Fraile et al., 2008; Lombard et al., 2011; Roy et al., 2015).

**Author contribution**

CB conceived and devised the project and methods, carried out the lab work and prepared the manuscript. KFD and BTN advised on methodology. ADR, CVD and JF collected planktonic foraminifera. AF contributed to next–generation sequencing and analysis. MW contributed to analysis of *Synechococcus* DNA. CB and KFD wrote the manuscript with contributions from all co–authors.

**Competing interests**

The authors declare that they have no conflict of interest.

**Acknowledgments**

CB is a Daphne Jackson Fellow sponsored by NERC and the University of Edinburgh via the Daphne Jackson Trust. This work was also supported by NERC award to MW ref: NE/K015095/1. Field collections were supported by NSF grant
number OCE-1261519. The authors would like to sincerely thank Bärbel Hönisch for collecting foraminifera for this study in July/Aug 2013. Thanks are also extended to Edinburgh Genomics and Steve Mitchell and Dave Kelly of Edinburgh University for microscopy assistance.

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

**Figure Legends**

Figure 1. Map of the Californian coast showing Bodega Head and Santa Catalina Island sampling locations. The hydrography of the region is described fully in Sect. 2.1, whilst the direction of the two major coastal currents, the California Current and the Davidson Current, are illustrated here.

Figure 2. Fluorescence micrograph of a DAPI stained decalcified *G. bulloides* cell. (a) Diffuse autofluorescence can be observed throughout the cytoplasm. The black arrowhead denotes an example of the bright spots, 3-10μm in size, that are vacuoles containing condensed prey items. The very many ~1 μm diameter bright globular structures throughout the cell are consistent with the presence of bacteria. The white rectangle denotes the area magnified in Figure 2(b). (b) The white arrow heads highlight two of the many clustered ~1 μm diameter structures that are consistent with the presence of bacterial cells which can be seen throughout the cell.

Figure 3. Relative abundance of taxonomically assigned 16S rRNA gene sequences from bacteria and chloroplasts within the cytoplasm of six individual foraminifer specimens; five *G. bulloides* (BUL22, BUL23, BUL34, BUL36 and BUL37) and one *N. dutertrei* specimen (DUT55). Sequences are assigned to operational taxonomic units (OTUs) grouped at different levels of taxonomic classification (see key). For example 16S rRNA gene sequences assigned to OTUs of the genus *Synechococcus* are the most abundant within *G. bulloides* and are at the highest level of classification.



Figure 4. Transmission electron microscope images of *Synechococcus* cells inside *G. bulloides*. (a) A *Synechococcus* cell with characteristic polyhedral carboxysomes in the central region (white arrowhead) surrounded by thylakoid membranes. Scale bar is 0.2 μm. (b) Numerous *Synechococcus* cells inside a *G. bulloides* cell are observed in both the cytoplasm and vacuoles, examples are indicated by black arrowheads. (c) *Synechococcus* cell within a *G. bulloides* cell undergoing cell division as indicated by the presence of a constriction at the cell midpoint (white arrowhead). Scale bar is 1 μm.





Fig01





Fig02





Legend:
- Genus Synechococcus
- Class Alphaproteobacteria; Family Bradyrhizobiaceae
- Class Alphaproteobacteria; other
- Class Gammaproteobacteria
- Class Betaproteobacteria
- Classes Delta- and Epsilonproteobacteria
- Phylum Planctomycetes
- Phylum Firmicutes
- Phylum Actinobacteria
- Phylum Bacteriodetes
- Other bacteria
- Phylum Streptophyta chloroplast (green algae)
- Phylum Stramenopile chloroplast (includes diatoms and chrysophyte algae)
- Phylum Haptophyceae chloroplast (includes coccolithophores)
- Phylum Cercozoa chloroplast (diverse protists)

Fig03





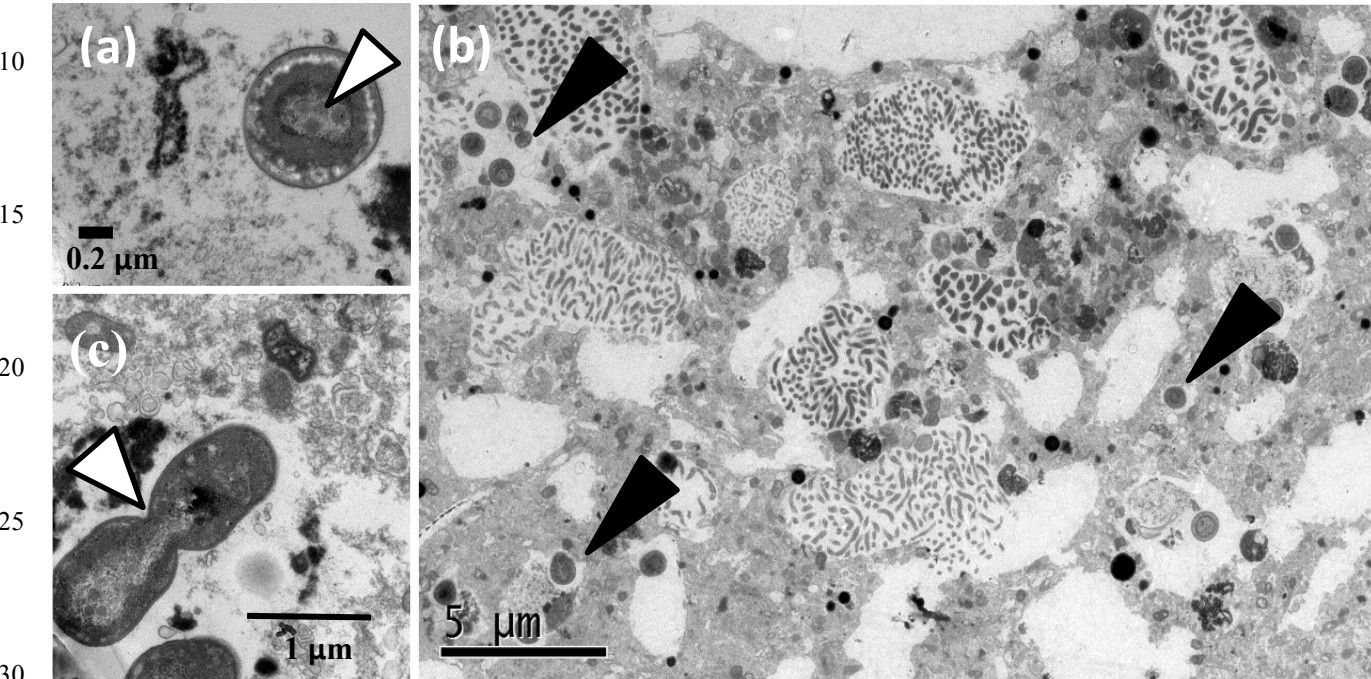

Fig04





Table 1. Sampling information and details of analyses performed for each planktonic foraminifer specimen collected

| Morphospecies | Sample ID | Sampling site | Sampling date | Co-ordinates | Sea Surface Temperature | Analysis |
|---|---|---|---|---|---|---|
| *G. bulloides* | BUL21 | Santa Catalina Island | July/Aug 2013 | 33.4°N, 118.4°W | 18°C -21.5°C | Control for fluorescence microscopy |
| *G. bulloides* | BUL24 | Santa Catalina Island | July/Aug 2013 | 33.4°N, 118.4°W | 18°C -21.5°C | DAPI staining and fluorescence microscopy |
| *G. bulloides* | BUL25 | Santa Catalina Island | July/Aug 2013 | 33.4°N, 118.4°W | 18°C -21.5°C | DAPI staining and fluorescence microscopy |
| *G. bulloides* | BUL26 | Santa Catalina Island | July/Aug 2013 | 33.4°N, 118.4°W | 18°C -21.5°C | DAPI staining and fluorescence microscopy |
| *G. bulloides* | BUL28 | Santa Catalina Island | July/Aug 2013 | 33.4°N, 118.4°W | 18°C -21.5°C | DAPI staining and fluorescence microscopy |
| *G. bulloides* | BUL29 | Santa Catalina Island | July/Aug 2013 | 33.4°N, 118.4°W | 18°C -21.5°C | DAPI staining and fluorescence microscopy |
| *G. bulloides* | BUL30 | Santa Catalina Island | July/Aug 2013 | 33.4°N, 118.4°W | 18°C -21.5°C | DAPI staining and fluorescence microscopy |
| *G. bulloides* | BUL32 | Santa Catalina Island | July/Aug 2013 | 33.4°N, 118.4°W | 18°C -21.5°C | TEM |
| *G. bulloides* | BUL34 | Santa Catalina Island | July/Aug 2013 | 33.4°N, 118.4°W | 18°C -21.5°C | Metabarcoding, genotyping and *Synechococcus* 16S and *rbcL* cloning and sequencing |
| *G. bulloides* | BUL36 | Santa Catalina Island | July/Aug 2013 | 33.4°N, 118.4°W | 18°C -21.5°C | Metabarcoding |
| *G. bulloides* | BUL37 | Santa Catalina Island | July/Aug 2013 | 33.4°N, 118.4°W | 18°C -21.5°C | Metabarcoding |
| *G. bulloides* | BUL39 | Santa Catalina Island | July/Aug 2013 | 33.4°N, 118.4°W | 18°C -21.5°C | TEM |
| *G. bulloides* | BUL69 | Santa Catalina Island | July/Aug 2013 | 33.4°N, 118.4°W | 18°C -21.5°C | TEM |
| *N. dutertrei* | DUT55 | Santa Catalina Island | July/Aug 2013 | 33.4°N, 118.4°W | 18°C -21.5°C | Metabarcoding and genotyping |
| *G. bulloides* | BUL04 | Bodega Head | Nov 2014 | 38.3°N, 123.0°W | 14°C-15°C | *Synechococcus* 16S and *rbcL*** |
| *G. bulloides* | BUL05 | Bodega Head | Nov 2014 | 38.3°N, 123.0°W | 14°C-15°C | *Synechococcus* 16S and *rbcL*** |
| *G. bulloides* | BUL13 | Bodega Head | Nov 2014 | 38.3°N, 123.0°W | 14°C-15°C | *Synechococcus* 16S and *rbcL*** |
| *G. bulloides* | BUL14 | Bodega Head | Nov 2014 | 38.3°N, 123.0°W | 14°C-15°C | *Synechococcus* 16S and *rbcL*** |
| *G. bulloides* | BUL15 | Bodega Head | Nov 2014 | 38.3°N, 123.0°W | 14°C-15°C | *Synechococcus* 16S and *rbcL*** |
| *G. bulloides* | BUL22 | Bodega Head | Nov 2014 | 38.3°N, 123.0°W | 14°C-15°C | Metabarcoding |
| *G. bulloides* | BUL23 | Bodega Head | Nov 2014 | 38.3°N, 123.0°W | 14°C-15°C | Metabarcoding |
| *G. bulloides* | BUL71 | Bodega Head | April 2015 | 38.3°N, 123.0°W | 10.5°C | TEM |
| *G. bulloides* | BUL73 | Bodega Head | April 2015 | 38.3°N, 123.0°W | 10.5°C | *Synechococcus* 16S and *rbcL*** |
| *G. bulloides* | BUL74 | Bodega Head | April 2015 | 38.3°N, 123.0°W | 10.5°C | *Synechococcus* 16S and *rbcL*** |
| *G. bulloides* | BUL82 | Bodega Head | April 2015 | 38.3°N, 123.0°W | 10.5°C | *Synechococcus* 16S and *rbcL*** |
| *G. bulloides* | BUL83 | Bodega Head | April 2015 | 38.3°N, 123.0°W | 10.5°C | *Synechococcus* 16S and *rbcL*** |
| *G. bulloides* | BUL84 | Bodega Head | April 2015 | 38.3°N, 123.0°W | 10.5°C | *Synechococcus* 16S and *rbcL*** |
| *G. bulloides* | BUL85 | Bodega Head | April 2015 | 38.3°N, 123.0°W | 10.5°C | *Synechococcus* 16S and *rbcL*** |
| *G. bulloides* | BUL86 | Bodega Head | April 2015 | 38.3°N, 123.0°W | 10.5°C | *Synechococcus* 16S and *rbcL*** |
| *G. bulloides* | BUL88 | Bodega Head | April 2015 | 38.3°N, 123.0°W | 10.5°C | *Synechococcus* 16S and *rbcL*** |

*PCR amplification of *Synechococcus* 16S rRNA gene and *rbcL* (RuBisCo large subunit)