# Peer review of "16S rRNA gene metabarcoding reveals a potential metabolic role for intracellular bacteria in a major marine planktonic calcifier (Foraminifera)."

_Biogeosciences, 2016_

## Short Comment (SC1) · 28 Nov 2016

16S rRNA gene metabarcoding and fluorescence microscopy can reveal the presence of bacteria in the cell (possibly digesting foods or endobionts), but cannot suggest ecological interaction between host and bacteria. How could you say the bacteria as endobionts?

The 16S rRNA gene metabarcodings were coming from amplicon sequences. Amplicon sequences are biased by primer, thus ratio of amplicon sequences did not mean the ratio of the bacteria community inside the cell. Also, the TEM image of possible

Synechococcus is difficult to observe thylakoid membrane. It is unclear for me to distinguish them as Synechococcus.
* * *

---

## Short Comment (SC2) · 5 Dec 2016

- The title is misleading as it is not supported by any result presented in this study, and should be changed.

- Synechococcus clade I also contains green-light specialists (e.g. strains ROS8604 or SYN20, see Pittera et al. 2014, ISME J, doi 10.1038/ismej.2013.228). Similarly, only two strains of clade IV have been characterized for pigmentation. The claim that all clade I and IV Synechococcus are chromatic acclimaters (discussion 4.1.5 and 4.2) is thus false for clade I and should be moderated for clade IV.

- In paragraph 4.1.1, you claim that PCR amplification of 16S RNA and rbcL "provides additional evidence that Synechococcus DNA [is] not grossly degraded by nucleases". However, you do amplify 16S marker gene for metabarcoding of bacteria that you claim are digested by the host. You thus use the same result to draw diametrically opposed conclusions.

- In paragraph 4.1.3, you observe 100% identity between 16S RNA and rbcL markers between "endobionts" and clade IV Synechococcus CC9902. However, strains CC9902 and BL107 share 100% and 98.9% nucleotide identity for these markers, yet share an average nucleotide identity of 91.3% for the core proteins (see Dufresne et al 2008, Genome Biology, doi 10.1186/gb-2008-9-5-r90), lower than the threshold value commonly used for bacterial species definition (94%). The conclusion "This strongly supports a strategy of horizontal rather than vertical transmission" should thus be moderated, as two strains exhibiting high degrees of similarity for these markers can be quite divergent for the rest of their genome: these markers thus provide evidence but are not sufficient to totally exclude genetic drift between internal and free-living Synechococcus.

---

## Referee Comment (RC1) · Anonymous Referee #1 · 13 Dec 2016

The manuscript by Bird et al. describes the first evidence for intracellular bacteria in planktonic foraminifera, marine protists. The authors use different approaches, microscopy, gene sequencing and metabarcoding, to investigate the presence and identity of bacteria within a planktonic foraminifer from two locations in the California Current. The main bacterial genus found is Synechococcus, which shows higher abundances inside the foraminiferal cell compared to the water column and seems to be actively dividing within the host. The authors thus consider it an endobiont within one genotype of the foraminifer Globigerina bulloides and they discuss potential metabolic roles of this bacterial genus in the association. The manuscript in general is well written

and presents the methods in great detail. The number of figures and the content of the supplements are suitable for the understanding of the MS. Being the first description of intracellular bacteria within a planktonic foraminiferal cell, I consider the manuscript of great importance for the understanding of the ecology of planktonic foraminifera, which further affects their application as proxies in paleontological studies. However, there are some issues regarding the interpretation of the data as well as some small technical issues, as I point out below, that need to be addressed by the authors.

General comments: Title: The title is too general and promises something that cannot be shown yet by the data. Metabarcoding only reveals the presence of the bacteria not their metabolic role. The title further seems to refer to all planktonic foraminifera in general, while the study only analyzes one genotype of one morphospecies.

Methodology: In part 4.1.4 of the discussion you mention the possibility of a primer bias against a certain group of bacteria introduced by the PCR based approach for the detection and identification of bacteria. I wonder if this may not be a more general problem in the study, making certain groups of bacteria appear more abundant than they actually are, since they are amplified more easily with the chosen primers than other groups. This issue needs to be discussed in the MS.

Discussion: Showing a metabolic role for intracellular bacteria in a eukaryote host sure is a difficult task. So for now this part should remain rather speculative and not appear for example in the title, as mentioned above. Yet, I agree that referring to the bacteria as endobionts is legitimate, as this purely describes their presence within another organism. As mentioned in part 4.1.2, Synechococcus are present in deep-living benthic foraminifera as well as diatoms. In both cases photosynthesis would not play any role in the association between the bacteria and the hosts. I wonder if Synechococcus just uses the hosts as some kind of protection, more or less "infesting" them. In this regard, I am not sure how the authors conclude that G. bulloides actively and species-specifically takes up the bacteria from the water column. I think there are no data yet to show how the bacteria really end up in the foraminiferal cell.

Detailed comments: Abstract: Line 2: Maybe edit to: "This marine protist is commonly used in micropaleontological investigations..." Lines 4-5: The reasoning why the authors chose G. bulloides to search for bacteria symbiosis is not completely clear. What does "atypical geochemical shell signature" and "divergent ecology" mean?

Introduction: Page 4 Lines 18-20: This sentence needs clarification: "...by more than any other extant, surface-dwelling species...Such large deviations..." Written like this, the statement needs quantification on how large the deviations actually are. Line 31: Is there only this one genotype in the sampling area? If yes then paleontological analysis on that morphospecies from that area should not contain any noise due to genotypes as mentioned in the paragraph before. Page 5 Line 1: Globigerina bulloides: In general the genus name should be written out only once at the beginning of each chapter and then afterwards abbreviated. Line 3-4: "We demonstrate..." I don't think it is really demonstrated here that the bacteria are actively taken up by the foraminifera and I also don't think it can be said yet if the association is really SPECIES-specific.

Material and Methods: Page 5 Line 18: I think it would be helpful to put the sampling point for the bacteria analysis in the water column in Figure 1. Line 22: How do the dates chosen for sampling relate to the oceanography and the changes in foraminifera abundances? Line 26: "species level": I assume this refers to morphospecies as the genotypes (which seem to be the actual species) cannot be differentiated morphologically as you mentioned before. Page 6: Line 2: I wonder how it is possible to make sure that all external contaminants are removed. By putting the shell in RNALater I assume that also contaminant DNA gets preserved. How is it possible to separate the foraminifera cell from the contaminants? Line 9: Maybe mention here which genes were amplified. Page 7: Line 7: I am not sure I understand which samples were pooled for the sequencing. The different individuals? How are they told apart again later on?

Results: Page 9: Line 29: In Table S1 the N. dutertrei individual is called DUT59. Page 10: Line 4: I think it could be helpful to show the unstained G. bulloides in the supplementary files to have a comparison between stained and unstained images. Of

course, even better would be a comparison to a stained species without (or with less) bacteria (e.g. N. dutertrei) to see the difference. Page 12: Line 3: I wonder how reliable the comparison between the bacteria in the foraminifera and the water column really is as the water column data were not taken together with the foraminifera sampling. I think it is necessary to further comment on these bacteria data to show how stable they are over time and how reliable it is to assume they were still valid at the time of sampling.

Discussion: Page 15: Line 30: "...with the majority of OTUs (>97%)..."

References: In general species names must be in italics.

Figure 1: The zoomed-out map is very small. I suggest making it larger and enhancing the contrast of the colors to make it more useful.

[Figure]

---

## Referee Comment (RC2) · Anonymous Referee #2 · 14 Dec 2016

This MS is a challenging study to unveil bacterial endosymbiosis in planktonic foraminifera. The topic is interesting for a wide range of people, if the results of this study have certainly demonstrated the presence of endosymbionts and their metabolisms. Here, I would like to mention four major questions about the methods and results of this study.

(1) Amplicon sequencing does not exactly show the quantity of DNA in the cell. The authors used amplicons, which were amplified by PCR, for 16S rDNA sequencing. In this process, the primers never randomly attach to DNA molecules. Even though

16S amplicon shows high abundance of DNA sequences of the cyanobacteria (Result 3.3), this does not mean high abundance of cyanobacteria in the foraminiferal cell. A large variation (37–87%) of the abundances of the cyanobacterial DNA sequences possibly shows a bias of PCR amplification. The authors removed some contaminated sequences from the samples based on the three negative controls (Method 2.6.1). However, we still find the sequences of the class Alphaproteobacteria in Fig. 3. What are they? Are they contaminates, preys, or symbionts?

(2) The data of 16S rDNA sequences does not show "living" bacteria in the foraminiferal cell. The DNA fragments are highly remained in the cell, if the cell (foraminifer) takes bacteria through endocytosis. As mentioned in the method section 2.2, the samples were immediately put in the buffer after collection. In this case, foods were not digested in a planktonic foraminifer. If G. bulloides is a bacteria-feeder, the DNA fragments of foods could be detected in 16S rDNA sequencing. This hypothesis (bacteria-feeder) is also reasonable to explain why the components of 16S rDNA sequences without cyanobacteria were different among the specimens (Fig. 3).

(3) The images of DAPI and TEM are not enough to support the presences of "living" bacteria in the foraminiferal cell. As the image of DAPI (Fig. 2) was unclear, it's difficult for me to follow the description in the result section 3.2. Although the TEM image was clear, there is no explanation about cytoplasm. Which are vacuoles? Are there phago-somes? The TEM images indicated the cell structures, which have carboxysomes and thylakoid membranes, as a character of the cyanobacteria. However, these images were not enough to certify that Synechococcus are living in the foraminiferal cell. As the body size of bacteria is very small rather than foraminifera, they can be remained in the foraminiferal cell. At least, the authors will need to count the number of bacteria-like cells in a planktonic foraminifer and show how these bacterial cells are universal. Moreover, I recommend the authors to use the FISH (Fluorescence in situ hybridiza-tion) method for detecting the "living" bacterial cells.

(4) General characters of Synechococcus did not prove their metabolic functions in the

foraminiferal cell. Synechococcus generally take or use nitrogen, carbon, and phosphorus. Their uptakes are different depending on species or clades. The current data and discussion (4.2) have no evidence which Synechococcus clade (I to IV) take these elements. Moreover, the authors should investigate intracellular distributions and assimilations of these elements. Please see one good example: Nomaki et al. (2016) published in Frontiers in Microbiology.

Because of these reasons, the title of this MS is not supported by any results. Although this study may show the presence of bacteria in a planktonic foraminifer, it is open to question whether those bacteria are endosymbionts or not. If the authors won't show any additional information to answer the questions as mentioned above, the title, abstract, and main text should be modified to report finding bacterial cells in a planktonic foraminifer. In particular, please delete the descriptions concerning the carbon isotope of the foraminiferal shells, because no data is suggesting this topic.

Minor comments:

a) Abstract (lines 3-4): Not "unusual". G. bulloides is one of spinose species without algal symbionts.

b) Abstract (line 8): There is no direct and own data about the bacterial populations in the water column.

c) Abstract (line 16): This study does not show the presence of bacterial enobiont.

d) Abstract (lines 17-19): Please delete.

e) Keywords: "symbiosis", "endobiont", and "carbon isotope" are deleted.

f) Introduction: The first two paragraphs should be omitted. Instead of them, the authors need to describe how the other studies demonstrated the presence of symbionts in Foraminifera and/or other organisms.

g) 2.2 Sample collection: Samples were collected from the vertical net-towing or surface water. In this case, it is very difficult to compare the hosts with bacterial components in different depths.

h) 2.3 Decalcification: Have the authors decalcify and wash the cell for all specimens?

i) 2.6. DNA extraction: Why did the authors use only one comparison (N. dutretrei)?

j) Discussion 4.1, 4.1.4: Based on amplicon sequencing, it is difficult to discuss the "abundance" of bacteria. Please see my comment (1).

k) Discussion 4.1.3: The authors used "selective uptake" in line 28. It's wrong, because there is no evidence.

l) Discussion 4.1.4: A visual coloration of planktonic foraminiferal cytoplasm does not demonstrate the characters of Synechococcus clades.

m) Figure 2: Unfortunately, Fig. 2 was somehow incomplete. Especially, I cannot find black arrows in Fig. 2a.

---

## Author Response (AR1)

1. **Response to Short Comment 1.**

   1. *16S rRNA gene metabarcoding and fluorescence microscopy can reveal the presence of bacteria in the cell (possibly digesting foods or endobionts), but cannot suggest ecological interaction between host and bacteria. How could you say the bacteria as endobionts?*

Future work will elucidate the nature of the relationship between *Synechococcus* and *G. bulloides* and until a benefit is demonstrated to either party we have refrained from using the term symbiont. We feel that the term endobiont is wholly appropriate in this instance.

The definition of an endobiont is of an organism that lives either below a surface (such as a sea bed) or inside another organism. It does not imply (beneficial) ecological interactions (although interactions must occur at a molecular level). We have used the term endobiont as our evidence suggests that *Synechococcus* are alive inside the *G. bulloides* cell. *Synechococcus* cell counts in TEM images demonstrate large numbers of *Synechococcus* cells inside *G. bulloides,* and in addition, that 5% of these cells are going through cellular division, i.e. they are reproducing.

   2. *The 16S rRNA gene metabarcodings were coming from amplicon sequences. Amplicon sequences are biased by primer, thus ratio of amplicon sequences did not mean the ratio of the bacteria community inside the cell.*

The primer set used in this study is that designed and used by the Earth Microbiome Project (Gilbert et al., 2010). The biases in this primer set are well known and have recently been corrected for (Apprill et al., 2015; Walters et al., 2016; Parada et al., 2016). The primer set has been tested with mock communities (Parada et al., 2016) and compares well with FISH results (Apprill et al. 2015) giving a good representation of the bacterial assemblages targeted. The bias in this primer set does not include an over amplification of *Synechococcus.* Therefore we believe that the proportions of *Synechococcus* demonstrated by this method are accurate and taken with the TEM cell counts, do reflect the true proportions of *Synechococcus* within the *G. bulloides* cell.

   3. *Also, the TEM image of possible Synechococcus is difficult to observe thylakoid membrane. It is unclear for me to distinguish them as Synechococcus.*

The thylakoid membranes can be observed circling the periphery of the cell, but we acknowledge that the clarity of these is not perfect. However, the carboxysomes, only found in cyanobacteria, are very clear, and the cell division in a single plain is also obvious. Both are characteristic of *Synechococcus*. TRITC excitation of *G. bulloides* cells under fluorescence microscopy (see new uploaded image), also demonstrates the presence of phycoerythrin-containing bacteria throughout the cell. Phycoerythrin is a pigment characteristic of *Synechococcus* and therefore all the evidence points to these cells being *Synechococcus.*

**2.  Response to Short Comment 2**

The authors wish to thank T. Grebert for taking the time to read this manuscript and for his useful and constructive comments, which the authors have enjoyed discussing here.

> 1.  *The title is misleading as it is not supported by any result presented in this study, and should be changed.*

The authors accept this in full. Future work will determine the metabolic role of the cyanobacteria identified within the foraminiferal cell in this study. Therefore the authors acknowledge that the title should be amended to "16S rRNA gene metabarcoding reveals the presence of intracellular cyanobacteria in a major marine planktonic calcifier (Foraminifera)"

> 2.  *Synechococcus clade I also contains green-light specialists (e.g. strains ROS8604 or SYN20, see Pittera et al. 2014, ISME J, doi 10.1038/ismej.2013.228). Similarly, only two strains of clade IV have been characterized for pigmentation. The claim that all clade I and IV Synechococcus are chromatic acclimaters (discussion 4.1.5 and 4.2) is thus false for clade I and should be moderated for clade IV.*

Thank you for this valid point. The text in discussion sections 4.1.5 and 4.2 will be modified accordingly to more accurately state that some strains within clade I and those so far characterized for pigments in clade IV are chromatic adapters. The rest of the discussion regarding the potential benefit of chromatic adaption to the host, remains valid.

> 3.  *In paragraph 4.1.1, you claim that PCR amplification of 16S RNA and rbcL "provides additional evidence that Synechococcus DNA [is] not grossly degraded by nucleases". However, you do amplify 16S marker gene for metabarcoding of bacteria that you claim are digested by the host. You thus use the same result to draw diametrically opposed conclusions.*

This is an excellent point that does need clarification. DNA degradation in "dietary samples" limits the size of DNA fragments that can be successfully amplified. For investigation of prey items target fragments should be limited to ~100-250 bp. Whilst longer fragments can be utilised, they will limit the success of amplification, so that sequences will not always be obtained (see Pompanon et al., 2012 doi: 10.1111/j.1365-294X.2011.05403.x).

In our study using 16S rRNA gene metabarcoding, we have amplified a fragment of 253 bp which will therefore give us information not only on intact bacteria but also on those bacteria that have been phagocytosed for food. Since the diet of foraminifera is not wholly known this information is also of value.  TEM imaging has then enabled us to further discriminate between food and endobiont.

We also PCR amplified partial *Synechococcus* 16S rRNA genes of 422 bp from *G. bulloides* total DNA for cloning and Sanger sequencing. This is a longer fragment than ideal for identifying prey (i.e. >250 bp) suggesting that the *Synechococcus* DNA is more intact than might be expected if it were the DNA of prey bacteria. Therefore the authors consider this to be supporting evidence (not stand alone evidence) that *Synechococcus* are living cells, endobionts, and not prey bacteria. The term "not grossly degraded by nucleases" is used in the manuscript to avoid over exaggeration of the significance of this data.

The authors consider that a further sentence in section 4.1.1 regarding the sizes of target DNA in helping to discriminate between prey and endobiont bacteria will be a helpful addition to the discussion.

> 4.  *In paragraph 4.1.3, you observe 100% identity between 16S RNA and rbcL markers between "endobionts" and clade IV Synechococcus CC9902. However, strains CC9902 and BL107 share 100% and 98.9% nucleotide identity for these markers, yet share an average nucleotide identity of 91.3% for the core proteins (see Dufresne et al 2008, Genome Biology, doi 10.1186/gb-2008-9-5-r90), lower than the threshold value commonly used for bacterial species definition (94%). The conclusion "This strongly supports a strategy of horizontal rather than vertical transmission" should thus be moderated, as two strains exhibiting high degrees of similarity for these markers can be quite divergent for the rest of their genome: these markers thus provide evidence but are not sufficient to totally exclude genetic drift between internal and free-living Synechococcus.*

Unlike in the example given above, both the 16S rRNA gene and the *rbcL* gene amplified and cloned from total *G. bulloides* DNA gave 100% identity to *Synechococcus* CC9902. However, we accept this point and will moderate our conclusion in section 4.1.3 accordingly with the caveat that there is a high degree of diversity among strains seemingly closely related through analysis of 16S and *rbcL* phylogenies.

**3.   Response to Referee #1**

The authors sincerely thank the referee for their time and constructive comments regarding this manuscript. We discuss those comments further below.

**General comments:**

> 1.  *Title: The title is too general and promises something that cannot be shown yet by the data. Metabarcoding only reveals the presence of the bacteria not their metabolic role. The title further seems to refer to all planktonic foraminifera in general, while the study only analyzes one genotype of one morphospecies.*

The authors fully accept that the metabolic role of *Synechococcus* has not been established and will remove this from the title, and will be more specific in referring to the foraminifera.

[Changes to manuscript] Title changed to: 16S rRNA gene metabarcoding reveals the presence of intracellular cyanobacteria in a major marine calcifier, *G. bulloides* (planktonic foraminifera).

> 2.  *Methodology: In part 4.1.4 of the discussion you mention the possibility of a primer bias against a certain group of bacteria introduced by the PCR based approach for the detection and identification of bacteria. I wonder if this may not be a more general problem in the study, making certain groups of bacteria appear more abundant than they actually are, since they are amplified more easily with the chosen primers than other groups. This issue needs to be discussed in the MS.*

We agree entirely with the referee that primer bias in this method needs to be addressed in the manuscript. So as not to repeat our comments on this point, we refer readers to both the "reply to SC1" document and to part (1) of "response to referee #2" on the discussion forum with regard to the bias in the primer set used in this study.

[Changes to manuscript] In Section 2.6 we have added a fuller description of the primer set and the known biases, and discuss a lack of amplification bias towards Synechococcus in Section 4.1.4.

3. *Discussion: Showing a metabolic role for intracellular bacteria in a eukaryote host sure is a difficult task. So for now this part should remain rather speculative and not appear for example in the title, as mentioned above. Yet, I agree that referring to the bacteria as endobionts is legitimate, as this purely describes their presence within another organism. As mentioned in part 4.1.2, Synechococcus are present in deep living benthic foraminifera as well as diatoms. In both cases photosynthesis would not play any role in the association between the bacteria and the hosts. I wonder if Synechococcus just uses the hosts as some kind of protection, more or less "infesting" them. In this regard, I am not sure how the authors conclude that G. bulloides actively and species-specifically takes up the bacteria from the water column. I think there are no data yet to show how the bacteria really end up in the foraminiferal cell.*

The authors acknowledge that any metabolic interactions between the two parties have not yet been demonstrated and we have altered the title accordingly. We consider Section 4.2 to be a speculative discussion on the potential benefits to either party in agreement with the request of the referee.

We agree with the referee that *Synechococcus* are endobionts, but that there is currently no data to show how large numbers of the cyanobacteria accumulate in the foraminiferal cell. For example does the *Synechococcus* population exist purely via cell division of a small number of cyanobacteria entering the cell, or does the foraminifera phagocytose many *Synechococcus* cells and actively maintain the population? Or are the *Synechococcus* cells able to avoid digestion unlike other phagocytosed bacteria, and how? So it is yet to be determined whether host or endobiont instigates this relationship.

However what is known is that there is selectivity in this process, as to date *G. bulloides* Type IId is the only planktonic foraminifera known to house *Synechococcus*, and we have also shown that *G. bulloides* Type IId cells retain only *Synechococcus* rather than all ingested cells, so rendering it a highly specific relationship. Never-the-less we will modify our conclusions and use more passive terms in describing the relationship.

[Changes to manuscript] In the introductory paragraph of the discussion (section 4.1) the sentence "However, we do have strong evidence that *G. bulloides* Type IId contains large numbers of the photoautotrophic picocyanobacterium, *Synechococcus,* and that they are preferentially taken up from the water column by *G. bulloides* and concentrated within the host cytoplasm" has been removed in favour of "However, we do have strong evidence that *G. bulloides* Type IId contains large numbers of the photoautotrophic picocyanobacterium, *Synechococcus*. *Synechococcus* accumulates within the host cytoplasm in abundances far greater than those found in bloom conditions in the California Bight or in other foraminifera species investigated. How this association occurs is unclear,……"

This still stresses the unparalleled numbers of *Synechococcus* observed within the cell but avoids talking of the "active selectivity" objected to by the referee.

The title of 4.1.3 is changed to "*Synechococcus* cells accumulate in the *G. bulloides* cytoplasm" to remove reference to *Synechococcus* being specifically taken up from the water column. The term "selectively accumulated" is replaced by "accumulate" in the first sentence and the word "selective" is removed from line 28.

**Detailed comments**:

1. *Abstract: Line 2: Maybe edit to: "This marine protist is commonly used in micropaleontological investigations. . ."*

[Changes to manuscript] Line 2 is amended as suggested above

2. *Abstract: Lines 4-5: The reasoning why the authors chose G. bulloides to search for bacteria symbiosis is not completely clear. What does "atypical geochemical shell signature" and "divergent ecology" mean?*

[Changes to manuscript] In order to bring further clarity to the use of *G. bulloides* for this study we have altered lines 4-5, removing the words atypical and divergent accordingly, and instead briefly describing the atypical geochemistry and ecology.

3. *Introduction: Page 4: Lines 18-20: This sentence needs clarification: ". . .by more than any other extant, surface-dwelling species. . .Such large deviations. . ." Written like this, the statement needs quantification on how large the deviations actually are.*

A single quantification is not appropriate here. The deviations from predicted values will vary by study and oceanographic location since the oceanographic setting likely affects the offsets from predicted values (e.g. if the carbonate ion concentration is different in two ocean basins, the carbon isotope offsets from predicted equilibrium will be different even at the same depth, temperature, etc.). For quantification, the reader should look to the cited studies.

[Changes to manuscript] Change in lines 18-20: The authors have re-worded these sentences to emphasise the point that *G. bulloides* precipitates its shell out of equilibrium with respect to both carbon and oxygen isotopes rather than focussing on the magnitude of that offset.

4. *Introduction: Page 4: Line 31: Is there only this one genotype in the sampling area? If yes then paleontological analysis on that morphospecies from that area should not contain any noise due to genotypes as mentioned in the paragraph before.*

There is only one extant genotype identified so far in this region (with the exception of a single individual considered to be potential contamination), and hence noise due to genotype variation is unlikely to occur here, rendering the geochemical calibrations from this area robust. The *G. bulloides* Type IId genotype has not, so far, been found elsewhere and as such the calibration for this genotype may not be appropriate for other genotypes. Understanding the ecology of each genotype and genotype-specific calibrations will be necessary across the morphospecies since mixed populations do exist that harbour different geochemical signatures.

[Changes to manuscript] Introduction: Line 31: Addition of phrase to confirm the presence of only one known genotype here to date. In addition a reference has been added to support the statement of different genotypes harbouring different geochemical signatures.

5. *Introduction: Page 5: Line 1: Globigerina bulloides: In general the genus name should be written out only once at the beginning of each chapter and then afterwards abbreviated.*

[Changes to manuscript] *Globigerina bulloides* has been changed to *G. bulloides*

6. *Introduction: Page 5: Line 3-4: "We demonstrate. . ." I don't think it is really demonstrated here that the bacteria are actively taken up by the foraminifera and I also don't think it can be said yet if the association is really SPECIES-specific.*

The authors acknowledge this point.

[Changes to manuscript] The sentence has been changed accordingly to "We demonstrate that intact viable cells of the picocyanobacteria *Synechococcus* accumulate in the cytoplasm of *G. bulloides* Type IId, that these cells are likely to be taken up from the water column and that *Synechococcus* should be considered an endobiont of *G. bulloides* Type IId."

7. *Material and Methods: Page 5: Line 18: I think it would be helpful to put the sampling point for the bacteria analysis in the water column in Figure 1. Also Figure 1: The zoomed-out map is very small. I suggest making it larger and enhancing the contrast of the colors to make it more useful.*

[Changes to manuscript] Figure 1 is amended as suggested.

8. *Materials and Methods: Page 5: Line 22: How do the dates chosen for sampling relate to the oceanography and the changes in foraminifera abundances?*

The authors wish to thank the referee for this valid point.

[Changes to manuscript] Additional information is added to Section 2.2 highlighting the specific oceanographic conditions at the time of collection.

9. *Materials and Methods: Page 5: Line 26: "species level": I assume this refers to morphospecies as the genotypes (which seem to be the actual species) cannot be differentiated morphologically as you mentioned before.*

[Changes to manuscript] species level altered to morphospecies level.

10. *Materials and Methods: Page 6: Line 2: I wonder how it is possible to make sure that all external contaminants are removed. By putting the shell in RNALater I assume that also contaminant DNA gets preserved. How is it possible to separate the foraminifera cell from the contaminants?*

The RNALater preserves all cells in stasis such that they do not lyse. Therefore any bacterial cells associated with the shell will remain intact and as the shell dissolves, those bacteria will be suspended in the RNALater. The washing steps are designed to remove as many of these bacteria as possible before the foram cell is transferred to DOC buffer for cell lysis, but we acknowledge that some contamination will remain.

To test our method we used a benthic foraminifera known to feed on diatoms and perform kleptoplastidy, an *Elphidium* species (genetic Type S4; Darling et al., 2016, DOI: 10.1016/j.marmicro.2016.09.001). Because this foraminifera contains chloroplasts these should be abundantly present in the 16S rRNA gene profile in metabarcoding, and their abundance should be proportional to the numbers of bacteria associated with the foraminifera. We performed metabarcoding on four individuals. Three of these had their shells dissolved via RNALater and were washed according to the method described in this manuscript. A final individual was placed directly into DOC buffer for DNA extraction for comparison. The metabarcoding results were conclusive. The individuals that had been through the dissolution and washing steps contained average abundances of 85%:15% chloroplast:bacteria 16S sequences and the individual crushed directly in DOC buffer contained 8%:92% chloroplast:bacteria 16S sequences. The washing method used in this study therefore removes considerable amounts of bacterial contaminants from the shell.

[Change to manuscript] A sentence has been added to describe preliminary washing experiments with the benthic kleptoplastic foraminifera, *Elphidium* species, and to refer readers to an in preparation manuscript.

*11. Materials and Methods: Page 6: Line 9: Maybe mention here which genes were amplified.*

[Changes to manuscript] "PCR amplification "of the foraminiferal 18S rRNA gene"" has been added.

*12. Materials and Methods: Page 7: Line 7: I am not sure I understand which samples were pooled for the sequencing. The different individuals? How are they told apart again later on?*

[Changes to manuscript] Section 2.6 has been reworded where necessary to make more clear which samples were pooled and to emphasize the use of a barcoded primer series which gives each sample a unique and identifiable tag enabling identification (demultiplexing) after the sequencing process.

*13. Results: Page 9: Line 29: In Table S1 the N. dutertrei individual is called DUT59.*

We apologise for this error.

[Changes to manuscript] This sample has been relabelled DUT55 in accordance with the specimen used and referred to throughout this study.

*14. Results: Page 10: Line 4: I think it could be helpful to show the unstained G. bulloides in the supplementary files to have a comparison between stained and unstained images. Of course, even better would be a comparison to a stained species without (or with less) bacteria (e.g. N. dutertrei) to see the difference.*

The authors thank the referee for this good suggestion. We are able to add an unstained *G. bulloides* fluorescence micrograph to the supplementary figures. However DAPI stained *N. dutertrei* images will be available in a separate manuscript.

[Changes to manuscript] Addition of supplementary image showing an unstained (no DAPI) *G. bulloides* cell under the DAPI filter set, labelled supplementary figure 1 and subsequent supplementary figure numbers changed sequentially.

15.   *Results: Page 12: Line 3: I wonder how reliable the comparison between the bacteria in the foraminifera and the water column really is as the water column data were not taken together with the foraminifera sampling. I think it is necessary to further comment on these bacteria data to show how stable they are over time and how reliable it is to assume they were still valid at the time of sampling.*

The concentration ranges and fluxes of *Synechococcus* across the world's oceans have been very well established over many years, and certainly it is particularly well monitored in the California Bight. Therefore a comparison between the bacteria in the foraminifera and the water column is entirely reliable. In this area maximum bloom concentrations (not always reached annually) have been measured at $6\times10^5$ cells per ml and hence our data which suggests concentrations of more than $1\times10^9$ cells per ml within the foraminifera are of huge significance.

[Changes to manuscript] In section 3.4 the well-established concentration range of *Synechococcus,* including bloom concentrations in the local area is highlighted more intensely.

16.   *Discussion: Page 15: Line 30: ". . .with the majority of OTUs (>97%). . ."*

[Changes to manuscript] Missing bracket has been inserted.

17. *References: In general species names must be in italics.*

[Changes to manuscript] All species names have been changed to Italics where necessary.

**4.   Response to Referee #2**

The authors wish to thank referee #2 for their time and contribution to this enjoyable discussion, and address her/his points below.

**General comments:**

*1.*   *Amplicon sequencing does not exactly show the quantity of DNA in the cell. The authors used amplicons, which were amplified by PCR, for 16S rDNA sequencing. In this process, the primers never randomly attach to DNA molecules. Even though 16S amplicon shows high abundance of DNA sequences of the cyanobacteria (Result 3.3), this does not mean high abundance of cyanobacteria in the foraminiferal cell. A large variation (37–87%) of the abundances of the cyanobacterial DNA sequences possibly shows a bias of PCR amplification. The authors removed some contaminated sequences from the samples based on the three negative controls (Method 2.6.1). However, we still find the sequences of the class Alphaproteobacteria in Fig. 3. What are they? Are they contaminates, preys, or symbionts?*

The authors agree with referee #2 that amplicon sequencing is not a quantitative method and hence we have presented our data as proportionality bar charts and not absolute numbers. However, the next-generation sequencing methods employed in

this study are currently routinely used to assess bacterial populations in a variety of environments. We accept that primer bias is a general and acknowledged weakness in any amplification procedure but it is not a weakness specific to our study *per se*.

As discussed with Yoshiyuki Ishitani, author of SC1 (many thanks to this contributor), the primer set employed in this study is that designed and used by the Earth Microbiome Project (Gilbert et al., 2010). The biases in this primer set are well known and have recently been corrected for (Apprill et al., 2015; Walters et al., 2016; Parada et al., 2016). The primer set has been tested with communities of known species composition (Parada et al., 2016) and compares well with FISH results (Apprill et al. 2015) giving a good representation of the bacterial assemblages targeted. The bias in this primer set does not include an over-amplification of *Synechococcus*. This is further demonstrated in the second foraminifera species presented in this study, *N. dutertrei*, where *Synechococcus* sequences were present but in a very low proportion. There is, therefore, no evidence to support the implication by referee #2 that this primer set is biased in favour of *Synechococcus*.

Instead, we suggest that the variation in the proportion of *Synechococcus* OTUs (37%-87%) in the dataset is exactly what we would expect given the sporadic nature of predation. Depending on whether a *G. bulloides* individual has recently phagocytosed bacteria or not at the point of sampling, will shift the proportions of "other" bacteria compared to the *Synechococcus* endobiont. See section 3.3 in the manuscript where this is stated, and also referee #2 comments in part 2 below where she/he agrees with this hypothesis.

The Alphaproteobacteria are a class of bacteria made up of a number of orders and families which encompass hundreds of species. One of the families of the Alphaproteobacteria, family *Bradyrhizobiaceae* is represented in our dataset by a large number of OTUs (OTUs are separated by a >3% difference in the DNA sequence). One of these OTUs was removed because it was the major contaminating OTU, as were a further 8 OTUs from the class Alphaproteobacteria (see section 2.6.1). The other OTUs within the class Alphaproteobacteria (including OTUs of the *Bradyrhizobiaceae* family) were not contaminants, i.e. they were not significantly amplified within the three controls and therefore they remained in the dataset and are represented in Figure 3. These Alphaproteobacteria along with all other bacteria are considered to be food because we were unable to observe any intact bacteria other than *Synechococcus* in TEM imaging.

[Changes in manuscript] In Section 2.6 we have added a fuller description of the primer set and the known biases, and discuss a lack of amplification bias towards Synechococcus in Section 4.1.4. We also make clear that we interpret the remaining Alphaproteobacteria OTUs as food items for the reasons laid out above via small additions to Section 4.3.

2. *The data of 16S rDNA sequences does not show "living" bacteria in the foraminiferal cell. The DNA fragments are highly remained in the cell, if the cell (foraminifer) takes bacteria through endocytosis. As mentioned in the method section 2.2, the samples were immediately put in the buffer after collection. In this case, foods were not digested in a planktonic foraminifer. If G. bulloides is a bacteria-feeder, the DNA fragments of foods could be detected in 16S rDNA sequencing. This hypothesis (bacteria-feeder) is also reasonable to explain why the components of 16S rDNA sequences without cyanobacteria were different among the specimens (Fig. 3).*

We agree with referee #2 that 16S rRNA sequences are not conclusive of "living" bacteria, and that our method of transferring samples to RNALater after collection would indeed mean that recently phagocytosed bacteria would not be completely digested. We refer back to comment 3 in our response to SC2 on this discussion forum: It has been demonstrated that DNA degradation in "dietary samples" limits the size of DNA fragments that can be successfully amplified. For investigation of prey items target fragments should be limited to ~100-250 bp. Whilst longer fragments can be utilised, they will limit the success of amplification, so that sequences will not always be obtained (see Pompanon et al., 2012 doi: 10.1111/j.1365-294X.2011.05403.x).

In our study using 16S rRNA gene metabarcoding, we have amplified a fragment of 253 bp which will therefore give us information, as referee #2 rightly points out, not only on intact bacteria but also on those bacteria that have been phagocytosed for food. Since the diet of foraminifera is not wholly known this information is also of value. TEM imaging has then enabled us to further discriminate between food and endobiont. Observations of intact and dividing *Synechococcus* cells, and of no other intact bacteria, demonstrates that the 16S rRNA genes amplified from other groups of bacteria cannot belong to an endobiont, and are most likely therefore, to be ingested prey.

We also PCR amplified partial *Synechococcus* 16S rRNA genes of 422 bp from *G. bulloides* total DNA for cloning and Sanger sequencing. This is a longer fragment than ideal for identifying prey (i.e. >250 bp) suggesting that the *Synechococcus* DNA is more intact than might be expected if it were the DNA of prey bacteria. Therefore the authors consider this to be supporting evidence (not stand alone evidence) that *Synechococcus* are living cells, endobionts, and not prey bacteria. The term "not grossly degraded by nucleases" is used in the manuscript to avoid over exaggeration of the significance of this data.

[Changes in manuscript] The authors consider that additional sentences in section 4.1.1 regarding the sizes of target DNA in helping to discriminate between prey and endobiont bacteria will be a helpful addition to the discussion.

> 3. *The images of DAPI and TEM are not enough to support the presences of "living" bacteria in the foraminiferal cell. As the image of DAPI (Fig. 2) was unclear, it's difficult for me to follow the description in the result section 3.2. Although the TEM image was clear, there is no explanation about cytoplasm. Which are vacuoles? Are there phagosomes? The TEM images indicated the cell structures, which have carboxysomes and thylakoid membranes, as a character of the cyanobacteria. However, these images were not enough to certify that Synechococcus are living in the foraminiferal cell. As the body size of bacteria is very small rather than foraminifera, they can be remained in the foraminiferal cell. At least, the authors will need to count the number of bacteria-like cells in a planktonic foraminifer and show how these bacterial cells are universal. Moreover, I recommend the authors to use the FISH (Fluorescence in situ hybridization) method for detecting the "living" bacterial cells.*

DAPI: The authors understand that the unclear nature of the DAPI images was as a result of their failure to upload properly in the first instance, an issue which has now been rectified. The DAPI images demonstrate the presence of thousands of bacteria within the host *G. bulloides* cell. The authors acknowledge that a comparison with an unstained (no DAPI) *G. bulloides* cell (suggested by referee #1) would be of benefit and will add such an image to the supplementary material. In addition, we will add an additional figure to the main manuscript of a *G. bulloides* cell observed with a TRITC filter set

which excites phycoerythrin, a highly labile pigment characteristic of *Synechococcus*. It can be observed confined in bacteria-sized bright spots right across the *G. bulloides* cell. (Such an image was uploaded in reply to SC1 and can therefore be assessed by the editor and interested parties). This confirms that these multitude of bacteria are indeed *Synechococcus* and that their cell membranes are intact. Water soluble phycoerythrin would very rapidly diffuse into the surrounding aqueous milieu if the cell membrane was compromised (see section 4.1.1).

TEM: The authors acknowledge a need for more labelling on Figure 4. TEM images. We feel justified however in labelling the cyanobacteria in these images as *Synechococcus*. The genus *Synechococcus* includes all unicellular cyanobacteria of the order *Chroococcales* that lack a laminated sheath but possess thylakoids, carboxysomes and divide by binary fission in a single plane (Rippka et al., 1979) as demonstrated in the TEM image. Other names have been used for individual species but not all (e.g. Microcystis) have the authority of the Bacteriological Code.

The authors disagree with referee #2 that the evidence provided does not indicate living *Synechococcus*. The authors assert that since 5% of all cyanobacterial cells counted within the *G. bulloides* cell were dividing, that this is strong support for a living endobiont, particularly given that cyanobacterial cells are within the cytoplasm and not within vacuoles. We believe that the use of the term endobiont as opposed to symbiont is wholly appropriate given the evidence. We refrain from using the term symbiont (or endosymbiont), since, as referee #2 quite rightly points out, we have not yet demonstrated a benefit to either party.

FISH: As suggested by referee #2, the authors did investigate the possibility of performing FISH, as an excellent method for determining living endobionts and their metabolic activity. However, in our hands the unparalleled foraminiferal cell autoflourescence observed under fluorescence microscopy, would have drowned out the signal generated by FISH, or indeed CARD-FISH. Under such unusual circumstances, therefore, FISH was considered an unsuitable method at this time for this organism.

As referee #2 suggests, we did in fact perform cell counts, but rather than using FISH and a fluorescence micrograph, we used TEM imaging. We calculated an average *Synechococcus* cell concentration up to 4 orders of magnitude greater in *G. bulloides* compared to concentrations known in the water column in this area (section 3.4). This high count is borne out by Fluorescence microscopy, particularly using TRITC (new figure to be added).

[Changes in manuscript] Addition of supplementary image (suggested by referee #1) showing an unstained (no DAPI) *G. bulloides* cell under the DAPI filter set to be labelled supplementary figure 1 and subsequent supplementary figure numbers to be changed. Addition of an extra figure to be labelled Fig. 3 (subsequent figure numbers to be changed) of a *G. bulloides* cell under the TRITC filter set demonstrating the fluorescence of the pigment Phycoerythrin which is characteristic of *Synechococcus*. Additional labelling on Figure 4 TEM images (which will be relabelled Figure 5).

4. *General characters of Synechococcus did not prove their metabolic functions in foraminiferal cell. Synechococcus generally take or use nitrogen, carbon, and phosphorus. Their uptakes are different depending on species or clades. The current data and discussion (4.2) have no evidence which Synechococcus clade (I to IV) take these elements.*

*Moreover, the authors should investigate intracellular distributions and assimilations of these elements. Please see one good example: Nomaki et al. (2016) published in Frontiers in Microbiology.*

*Because of these reasons, the title of this MS is not supported by any results. Although this study may show the presence of bacteria in a planktonic foraminifer, it is open to question whether those bacteria are endosymbionts or not. If the authors won't show any additional information to answer the questions as mentioned above, the title, abstract, and main text should be modified to report finding bacterial cells in a planktonic foraminifer. In particular, please delete the descriptions concerning the carbon isotope of the foraminiferal shells, because no data is suggesting this topic.*

The authors acknowledge that the current study does not present data on the metabolic interactions between *G. bulloides* and *Synechococcus*. Future work will elucidate the benefits to either party. Therefore we are happy to modify the title and moderate aspects of the manuscript accordingly. However, the data strongly support our conclusions that *Synechococcus* is an endobiont of *G. bulloides* Type IId. As a living endobiont, the respiration of *Synechococcus* is absolutely pertinent to the carbon isotopic ratio of the *G. bulloides* shell and demands discussion. Geochemical proxies based on *G. bulloides* are particularly important for palaeoceanographic reconstructions because they provide a link between subtropical and high-latitude species. Bacterial respiration could contribute isotopically depleted respiratory carbon to the calcite shell, explaining part of the offset between measured and predicted d13C.

[Changes in manuscript] Title changed to: 16S rRNA gene metabarcoding reveals the presence of intracellular cyanobacteria in a major marine calcifier, *G. bulloides* (planktonic foraminifera).

**Minor comments:**

*1. Abstract (lines 3-4): Not "unusual". G. bulloides is one of spinose species without algal symbionts.*

We used the term "unusual" to refer to two qualities of *G. bulloides*. Firstly to its lack of algal symbionts. Unusual does not imply unique, but it does mean distinct: the distinction being that many other spp. have algal symbionts. Out of the 11 spinose species living in the photic zone whose symbiont status is reported by Hemleben et al. 1989, 3 lack symbionts, but one of these (*Globigerinella calida*) has since been reported to have symbionts. Only 5 of the 11 are paleoceanographically important and out of those, *G. bulloides* is the only one without symbionts, and hence is unusual. Secondly we used the term "unusual" in reference to *G. bulloides* shell geochemistry which is universally accepted as being so, and hence we are fully justified in using the term "unusual".

*2. Abstract (line 8): There is no direct and own data about the bacterial populations in the water column.*

This is correct, and the authors acknowledge that having our own data for comparison would be of value. Unfortunately it was not possible to collect such data during this work. However, the SPOT site has been studied extensively over 10 annual cycles. The microbial composition and seasonal variability are well established therefore giving a clear overview of the water column assemblage (compared to a handful of snap shot samples) with which to compare the *G. bulloides* assemblage.

[Changes in manuscript] Line 8 is modified to read "To investigate the ecological interactions between *G. bulloides* and marine bacteria,……"

>  3.  *Abstract (line 16): This study does not show the presence of bacterial enobiont.*

We have refrained from using the term symbiont as we would agree that since no mutual benefit, or indeed a benefit to one or the other partners has been demonstrated in this study the term symbiont should not be used. However all the data presented in this study points to *Synechococcus* living inside the *G. bulloides* cell, so the term endobiont is absolutely appropriate.

>  4.  *Abstract (lines 17-19): Please delete.*

The authors consider that, given that we have presented many convincing lines of argument to suggest that *Synechococcus* is an endobiont of *G. bulloides*, lines 17-19 are valid statements pertaining to the manuscript discussion.

Keywords: "symbiosis", "endobiont", and "carbon isotope" are deleted.

We accept that symbiosis should be deleted, but assert that "endobiont" and "carbon isotope" should remain, for reasons discussed above.

[Changes in manuscript] keyword "symbiosis" deleted

>  5.  *Introduction: The first two paragraphs should be omitted. Instead of them, the authors need to describe how the other studies demonstrated the presence of symbionts in Foraminifera and/or other organisms.*

Referee #2 makes an excellent suggestion to describe how other studies have demonstrated the presence of symbionts/endobionts and we thank them for it. However the authors feel that the first two paragraphs on the foraminiferal contribution to the carbonate flux and their importance to palaeoceanography is of significance and therefore needs to be introduced, giving weight to the need for investigation of these organisms.

[Changes in manuscript] The first two paragraphs are revised and shortened rather than omitted. A short new paragraph is inserted, as suggested, to describe how the presence of protist algal symbionts has been demonstrated.

>  6.  *Section 2.2 Sample collection: Samples were collected from the vertical net-towing or surface water. In this case, it is very difficult to compare the hosts with bacterial components in different depths.*

Off Santa Catalina Island, near the SPOT site, where the bacterial population in the water column is well documented (see point (b) above), the foraminifera were collected by scuba or net tows in the surface waters (<25m) and therefore these specimens can be compared with the well-documented bacterial composition of the water column.

At Bodega Head samples were collected by vertical net tows, but these are not close to the SPOT site and therefore the two *G. bulloides* specimens collected from here and used in metabarcoding were not directly compared with the water column assemblage. However, statistical analysis (Bray-Curtis and LEfSe analysis) showed no significant difference between the bacterial composition of *G. bulloides* individuals from Catalina and those from Bodega Head.

>  7.  *Section 2.3 Decalcification: Have the authors decalcify and wash the cell for all specimens?*

Yes, see section 2.3.

8. *Section 2.6. DNA extraction: Why did the authors use only one comparison (N. dutretrei)?*

*N. dutertrei* was used as a comparison because it was the species collected at the same time and location as the Santa Catalina *G. bulloides* and therefore provides a direct comparison to both validate the methods used here and highlight that there are differences between species and also consistency across the *G. bulloides* individuals.

5   [Changes in manuscript] A small addition to section 3.3 clarifies why *N. dutertrei* was used as a comparison.

9. *Discussion 4.1, 4.1.4: Based on amplicon sequencing, it is difficult to discuss the "abundance" of bacteria. Please see my comment (1).*

In the title of Section 4.1 we use the term "abundant" to refer to the *Synechococcus* endobiont. It is a statement not based on amplicon sequencing, but on our microscopy data and cell counts which clearly demonstrate that *Synechococcus* is indeed

10   abundant within the *G. bulloides* cell. We refer the editor particularly to the new figure (this will be figure 3. in the revised manuscript) highlighting phycoerythrin fluorescence. In Section 4.1.4 we would like to point out that the term "abundant" is preceded by the word "relative" when referring to OTUs (which are based on amplicon sequencing) thereby indicating the proportional, and not absolute, nature of the data.

10. *Discussion 4.1.3: The authors used "selective uptake" in line 28. It's wrong, because there is no evidence.*

15   The authors suggest that the evidence for selective uptake is the 4 orders of magnitude greater numbers of *Synechococcus* inside the *G. bulloides* cell compared to the highest numbers of *Synechococcus* found in the region. However the authors agree that it is not yet know how the *Synechococcus* arrive in the foraminiferal cell, whether the cyanobacterial population exists purely via cell division of a small number of bacteria entering the cell, or whether *Synechococcus* are phagocytosed in large numbers, and hence will modify this section accordingly.

20   [Changes in manuscript] The title of 4.1.3 is changed to "*Synechococcus* cells accumulate in the *G. bulloides* cytoplasm" to remove reference to *Synechococcus* being specifically taken up from the water column. The term "selectively accumulated" is replaced by "accumulate" in the first sentence and the word "selective" is removed from line 28. An additional sentence is added to stress that it is not yet known how the *Synechococcus* cells accumulate in the cytoplasm in order to move away from the concept of "selective uptake".

11. *Discussion 4.1.4: A visual coloration of planktonic foraminiferal cytoplasm does not demonstrate the characters of Synechococcus clades*

We assume that this is in reference to Discussion section 4.1.5. The referee is correct. The colouration of the foraminiferal cytoplasm isn't evidence of *Synechococcus* clades; the systematic evidence from both 16S and *rbcL* genes is: these are

30   sequences predominantly from Clade IV *Synechococcus*

[Changes to manuscript] An additional few words have been added to clarify that not all Clade IV *Synechococcus* have yet been characterised for pigments, which is further clarified in the following Section 4.2.

12. *Figure 2: Unfortunately, Fig. 2 was somehow incomplete. Especially, I cannot find black arrows in Fig. 2a*

The authors understand that this was a technical issue regarding the initial upload of the manuscript. This has since been rectified and Figure 2 is now complete.

**5. List of Changes to manuscript. Please note page and line numbers referred to are relevant to this document only.**

1. **Title.** Changed to: Cyanobacterial endobionts within a major marine, planktonic, calcifier (Globigerina bulloides, Foraminifera) revealed by 16S rRNA metabarcoding. (requested by all)
2. **Abstract. Page 19. Line 2.** Addition of the word microplaeontological (requested by ref#1)
3. **Abstract. Page 19. Lines 5-7.** Addition of a sentence to describe the "atypical" shell geochemistry and "divergent" ecology (requested by ref#1)
4. **Abstract. Page 19. Line 10.** Replacement of "bacterial populations in the water column" with "marine bacteria" (requested by ref#2)
5. **Introduction. Page 20-21.** An initial paragraph has been added regarding symbiosis in planktonic foraminifera, the original first two paragraphs describing the importance of planktonic foraminifera have then been revised and shortened.
6. **Introduction. Page 22. Line 23-24**. The authors have re-worded these sentences to emphasise the point that *G. bulloides* precipitates its shell out of equilibrium with respect to both carbon and oxygen isotopes rather than focussing on the magnitude of that offset. (requested by ref#1)
7. **Introduction**. **Page 23. Lines 2-10.** The term "species-specific" has been replaced with "genotype-specific" and sentences have been added to describe the genotypes in the region. In addition a reference has been added to support the statement of different genotypes harbouring different geochemical signatures (requested by ref#1).
8. **Introduction**. **Page 23. Line 11.** *Globigerina bulloides* has been changed to *G. bulloides* (ref#1)
9. **Introduction**. **Page 23. Lines 15-19.** These lines have been altered to avoid use of the term active and species-specific uptake (requested by ref#1 and #2).
10. **Materials and methods. Section 2.2. Page 24. Lines 4-8 and 14-15.** Addition of information regarding the specific oceanographic conditions at time of sample collection. (requested by ref#1)
11. **Materials and methods. Section 2.2. Page 24. Line 12.** Addition of the prefix "morpho" to "species" (requested by ref#1)
12. **Materials and methods. Section 2.3. Page 24. Lines 22-24.** Addition of s sentence describing the efficacy of the washing treatment. (requested by ref#1)
13. **Materials and methods. Section 2.4. Page 24. Line 27.** Addition of the gene amplified. (requested by ref#1)
14. **Materials and methods. Section 2.6. Pages 25-26.** This section has been reworded with a number of additions for further explanation of (i) the biases in the primer set used, (requested by refs#1 and 2 and SC1) and (ii) clarification on the pooling and demultiplexing of DNA samples (requested by ref#1).
15. **Materials and methods. Section 2.6.1. Page 26. Line 24.** Correction of "genus" Alphaproteobacteria to "class" Alphaproteobacteria.
16. **Results. Section 3.2. Page 29. Lines 2-3.** Insertion of reference to **"**Supplementary Figure S1" which is now an unstained *G. bulloides* cell observed under the DAPI filter set (requested by ref#1).
17. **Results. Section 3.2. Page 29. Line 11.** Insertion of reference to new figure 3, the *G. bulloides* cell under TRITC excitation.
18. **Results. Section 3.3. Page 29. Line 27.** Change Supplementary figure 1 to supplementary figure 2.
19. **Results. Section 3.3. Page 29. Lines 30-31.** Addition of phrases to describe the use of *N. dutertrei* as a comparison to *G. bulloides* (requested by ref#2).
20. **Results. Section 3.3. Page 29. Lines 31 and 32.** Change Fig. 3 to Fig. 4 due to additional figure.

21. **Results. Section 3.4. Page 30. Lines 29, 31 and 33.** Change to "Fig. 4a/b/c" to "Fig. 5a/b/c" due to additional figure.
22. **Results. Section 3.4. Page 31. Lines 6-13.** Additional phrases to stress that the concentrations of *Synechococcus* including bloom concentrations in the water column are well known and established (requested by ref#1).
23. **Results. Section 3.5. Page 31. Line 21.** Change Supplementary figure 2 to supplementary figure 3.
24. **Discussion. Section 4.1. Page 32. Line 11.** *Globigerina bulloides* is changed to *G. bulloides* as it has been referred to previously (requested by ref#1)
25. **Discussion. Section 4.1. Page 32.** Rephrasing of this paragraph to remove emphasis on selective/active uptake of *Synechococcus* by *G. bulloides* (requested by refs#1 and #2).
26. **Discussion. Section 4.1.1. Pages 32-33.** This section has been rephrased to discuss the size of target DNA in helping to discriminate between prey and endobiont bacteria (requested by SC2 and ref#2).
27. **Discussion. Section 4.1.1. Page 32. Lines 28, 29 and 30.** Figure numbers have been changed due to the new additional figure.
28. **Discussion. Section 4.1.1. Page 33. Line 9.** Reference to Figure 3 has been inserted.
29. **Discussion. Section 4.1.2. Page 33. Lines 18-19.** Addition of phrase "(identified as cyanobacteria by…) and a correction to the reference from "Seckbach" to Buck and Bernhard 2006. (Seckbach is the editor of the book containing the Buck and Bernhard paper now referenced).
30. **Discussion. Section 4.1.3. Pages 33-34.** The title of this section and aspects of this paragraph have been altered to remove emphasis on selective/active uptake (requested by refs #1 and #2)
31. **Discussion. Section 4.1.3. Page 34. Lines 23-24.** Addition of the sentence "Whilst there can be a high degree of diversity among strains seemingly closely related through *rbcL* and 16S rRNA gene phylogenies, this evidence supports….." to acknowledge that although it is likely that the phylogenies accurately depict the cladal groupings, they are not absolute (requested by SC2).
32. **Discussion. Section 4.1.4. Page 34. Line 28.** Removal of "very specific uptake" again to remove emphasis on selective/active uptake of bacteria b the foraminifer (change in emphasis requested by ref#1 and ref#2).
33. **Discussion. Section 4.1.4. Page 35. Lines 6-8.** Addition of a sentence regarding the lack of bias towards *Synechococcus* in the primer set used (requested by SC1, ref#1 and ref#2).
34. **Discussion. Section 4.1.4. Page 35. Line 15.** The term species-specific is changed for genotype-specific.
35. **Discussion. Section 4.1.4. Page 35. Line 17.** Fig. 3 changed to Fig. 4
36. **Discussion. Section 4.1.4. Page 35. Line 18.** Missing bracket inserted (ref#1).
37. **Discussion. Section 4.1.4. Page 35. Line 20.** "species-specific" is replaced with "morphospecies/genotype-specific" as it is a comparison between the morphospecies of *N. dutertrei* and *G. bulloides* and also between the specific genotypes.
38. **Discussion. Section 4.1.5. Page 35, Line 23 –33.** This section has been reworded for clarity. A sentence has been added to confirm that the discussion pertains to *G.* bulloides collected in this area only. Also a sentence has been added to clarify that not all clade IV *Synechococcus* have yet been characterised for pigments (requested by SC2).
39. **Discussion. Section 4.2. Page 36. Lines 10-11.** Sentence re-phrased to clarify that not all clade IV *Synechococcus* have yet been characterised for pigments and that some clade I and those characterised from clade IV are chromatic adapters (requested by SC2).
40. **Discussion Section 4.3. Page 37. Lines 4-5 and 6-7.** Information regarding the oceanography at the time of sampling is added to give further reasons for the differences in the specimens from the different sampling locations (requested by ref#1)
41. **Discussion Section 4.3. Page 37. Lines 13 and 20.** Fig. 3 is changed to Fig. 4.
42. **Discussion Section 4.3. Page 37. Lines 17-18.** Addition of bracketed phrase confirming that some OTUs of the family Bradyrhizobiaceae were not contaminating OTUs and remained in the data set (requested by ref#2).
43. **References.** Species names have been put into italics where necessary (ref#2).

44. **References.** Deletion and addition of various references resulting from changes to the introduction, methods sections 2.2 and 2.6 and a correction in section 4.1.2 outlined above.
45. **Figure 1. Page 57.** This has been modified with corresponding figure legend amendment **Page 56** (requested by ref#1)
46. **New Figure 3. Page 59.** This is a new addition for further clarity of the fluorescence microscope work, with new figure legend inserted **Page 56**.
47. **Old Figure 3. Page 60.** Bar chart of 16S rRNA gene OTU abundances and corresponding legend changed to Figure 4. **Page 56.**
48. **Old Figure 4 Page 61.** TEM image changed to Figure 5 and an additional label added highlighting the unusual fibrillary bodies with corresponding additions to the figure legend. **Page 56** (requested by ref#2).
49. **Table 1 Page 62.** Correction, DUT59 corrected to DUT55 (ref#1)
50. **New Supplementary Figure 1. In supplementary material document.** Addition of extra supplementary figure (new S1) with additional figure legend showing an unstained *G. bulloides*. All subsequent supplementary figure numbers are changed in sequence.

[revised manuscript text omitted]